# GraphPFN: A Prior-Data Fitted Graph Foundation Model

## Abstract

Graph foundation models face several fundamental challenges including transferability across datasets and data scarcity, which calls into question the feasibility of graph foundation models at all. However, despite similar challenges, the tabular domain has recently witnessed the emergence of the first successful foundation models such as TabPFNv2 or LimiX. Many of these models are based on the prior-data fitted networks (PFN) framework, in which models are pretrained on carefully designed synthetic datasets to make predictions in an in-context learning regime. Recently, G2T-FM has made the first step towards adopting PFNs for graph tasks, yet it is limited to hand-crafted features and was never pretrained on graph data. In this work, we make the next step by proposing GraphPFN, a PFN-based model designed and pretrained specifically for graphs. Following the PFN framework, we first design a prior distribution of synthetic attributed graphs by using a novel combination of multiple stochastic block models and a preferential attachment process for structure generation and graph-aware structured causal models for attribute generation. Then, we augment the tabular foundation model LimiX with attention-based graph neighborhood aggregation layers and train it on synthetic graphs sampled from our prior. On diverse real-world graph datasets with up to 50,000 nodes, GraphPFN shows strong in-context learning performance and achieves state-of-the-art results after finetuning, outperforming both G2T-FM and task-specific GNNs trained from scratch on most datasets. More broadly, we hope that GraphPFN shows the potential of PFN-based models for building graph foundation models.[1]

## 1 Introduction

In recent years, foundation models have significantly advanced the state of the art in domains such as natural language processing and computer vision. These models can learn from large unannotated datasets and generalize across a wide range of downstream tasks. Notable examples include large language models like BERT (Devlin et al., 2019) and GPT-3 (Brown et al., 2020) in natural language processing and the Vision Transformer (Dosovitskiy et al., 2020) and CLIP (Radford et al., 2021) in computer vision. These models often use self-supervised or unsupervised pretraining to learn rich, transferable representations. This approach has changed how models are built and used, removing the need for task-specific models and reducing dependence on large labeled datasets for every single task. Inspired by these successes, there is growing interest now in extending the foundation models methodology to other modalities, including graphs.

However, developing graph foundation models (GFMs) is much more challenging. Unlike text and images, graph data does not constitute *a single domain*. Instead, graphs are used to represent data from *different domains*, e.g., social networks (both virtual and real-world), information networks, transportation networks, co-purchasing networks, various physical, biological, or engineering systems, or even networks of abstract concepts. As a consequence, both the structure of a graph and its attributes (features and labels) may vary significantly across graph datasets and tasks. Moreover, the amount and diversity of the available graph data are significantly lower compared to computer vision and natural language processing. Taken together, these challenges call into question the feasibility of graph foundation models at all.

---

[1]Our code is available at https://anonymous.4open.science/r/640ebd34c8a0.

However, despite facing similar challenges, tabular machine learning has recently witnessed the emergence of the first successful tabular foundation models (TFMs) such as TabPFNv2 (Hollmann et al., 2025) or LimiX (Zhang et al., 2025). Many of these models are based on the framework of prior-data fitted networks (PFNs, Müller et al., 2021; Hollmann et al., 2023). PFNs are pretrained on diverse synthetic datasets drawn from a *prior* to approximate Bayesian inference and can make predictions in an in-context learning setting. Strong performance in the tabular domain suggests that PFNs offer a promising path towards building foundation models for both tabular and graph domains.

Recent works G2T-FM (Eremeev et al., 2025) and TAG (Hayler et al., 2025) have made the first step towards adopting PFNs for graph tasks by utilizing foundation models for tabular data to create graph foundation models. For this, they augment node features with graph-based information such as neighborhood-aggregated features or Laplacian positional encodings. This allows for transforming a graph node-level prediction problem into a tabular prediction problem and applying an existing tabular foundation model to this task. The resulting approach shows strong performance, but such models still depend heavily on hand-crafted features and lack large-scale pretraining on diverse graph data. As a result, they are limited in their ability to capture complex graph patterns.

In this work, we make the next step by proposing GraphPFN, a PFN-based model designed and pretrained specifically for graph data. Following the PFN framework, we pretrain GraphPFN on synthetic datasets drawn from a carefully designed graph prior. For generating graph structures, we propose an approach that combines multiple stochastic block models and augments them with a preferential-attachment process. We then generate graph-structure-dependent node attributes for our graphs by augmenting tabular structured causal models (SCMs, Hollmann et al., 2023; Qu et al., 2025) typical for tabular PFNs with message-passing mechanisms at random SCM nodes. This method allows us to efficiently generate millions of realistic and diverse synthetic graph datasets. Then, we initialize GraphPFN from the tabular foundation model LimiX (Zhang et al., 2025) and add an attention-based message-passing layer to each block of LimiX. This allows the model to learn complex graph-specific patterns while retaining its ability to handle diverse features and labels inherited from LimiX.

Our experiments show that on diverse real-world graph datasets with up to 50,000 nodes, GraphPFN achieves strong in-context learning performance, competitive with the best current models — well-tuned traditional GNNs (Kipf & Welling, 2017; Hamilton et al., 2017; Veličković et al., 2018; Shi et al., 2021) with improved architectures (Platonov et al., 2023b) and G2T-FM (Eremeev et al., 2025). Furthermore, after finetuning, GraphPFN outperforms all other approaches, setting a new state-of-the-art for the considered datasets.

Overall, our main contributions can be summarized as follows:

- We propose GraphPFN, which is, to the best of our knowledge, the first publicly available PFN-based model designed and pretrained specifically for graph data.

- We introduce a novel graph prior for efficient generation of realistic synthetic attributed graphs.

- We demonstrate that finetuned GraphPFN outperforms both strong traditional GNN baselines and existing graph foundation models.

More generally, our study shows that pretraining graph-aware PFNs on synthetic graph datasets from a well-designed graph prior is a promising direction for building powerful and generalizable graph foundation models.

## 2 RELATED WORK

### 2.1 PRIOR-DATA FITTED NETWORKS

Prior-data fitted networks (PFNs) were first introduced by Müller et al. (2021). The main idea behind PFNs is to train models that can make predictions on previously unseen datasets in a single forward pass. These models leverage in-context learning (ICL): rather than updating model parameters for each new dataset, they use the context provided at inference time to adapt their predictions without additional training.

In the PFN framework, the input to the model consists of two parts: a set of training samples with their labels, called the *context*, and a set of test samples without labels, called the *query*. During a single forward pass, the model uses the context to make predictions for the query samples, thus performing ICL. In practice, PFNs are commonly implemented as Transformers with a specific attention mask (Hollmann et al., 2023; 2025): context (training) samples attend to all other context samples, while query (test) samples are only allowed to attend to context samples and not to each other. This structure ensures that predictions for each query are based solely on the training data.

PFNs are trained via pretraining on a large collection of synthetic datasets. To achieve this, one specifies a *prior* over supervised datasets, and the model is trained to perform ICL as described above, predicting labels for query samples given context samples. As shown by Müller et al. (2021), this procedure trains the network to approximate the posterior predictive distribution under the chosen prior, which provides the main theoretical motivation for the approach.

## 2.2 TABULAR FOUNDATION MODELS

In their pioneering work TabPFN (Hollmann et al., 2023) and its successor TabPFNv2 (Hollmann et al., 2025), the authors proposed to utilize the framework of prior-data fitted networks to create tabular foundation models and showed that such models can achieve strong results, competitive with other approaches (Erickson et al., 2025). Nowadays, TFMs have become an active research area, with several new methods released recently (Zhang & Robinson, 2025; Zhang et al., 2025). Some methods focus on scalability (Qu et al., 2025) or faster inference (Mueller et al., 2025), while others emphasize training on real-world datasets rather than synthetic tasks (Ma et al., 2024). Together, these works broaden the design space of tabular foundation models by trading off data sources, computational efficiency, and scalability.

## 2.3 GRAPH FOUNDATION MODELS

Similar to foundation models for tabular data, graph foundation models also face the challenge of handling datasets from diverse domains. A particularly difficult, yet crucial, aspect is managing the wide variety of node attributes (features and labels) present in different graphs. Early GFMs did not fully address this issue. They often relied on dimensionality reduction techniques such as principal component analysis or singular value decomposition (Xia & Huang, 2024; Zhao et al., 2024; Wang et al., 2025; Yu et al., 2025), or they restricted their focus to graphs where node attributes are all textual (Wang et al., 2024; He & Hooi, 2024; Liu et al., 2024).

More recent works, such as G2T-FM (Eremeev et al., 2025) and TAG (Hayler et al., 2025), have explored leveraging tabular foundation models to better address feature diversity in graph datasets. These approaches incorporate hand-crafted features, for example, neighborhood-aggregated features or Laplacian positional encodings, to effectively convert graph information into tabular features. Empirical results show that these methods achieve strong results, always significantly outperforming prior GFMs (Xia et al., 2024; Xia & Huang, 2024; Zhao et al., 2024; Finkelshtein et al., 2025; Zhao et al., 2025) and frequently outperforming well-tuned GNNs trained from scratch (Kipf & Welling, 2017; Hamilton et al., 2017; Veličković et al., 2018; Shi et al., 2021) with improved architectures (Platonov et al., 2023b), supporting the utility of employing tabular foundation models as a basis for learning on graph data.

## 3 GRAPHPFN

GraphPFN is a foundation model designed for in-context learning on graph-structured data. Inspired by recent advances in prior-data fitted networks (PFNs) for tabular data (Hollmann et al., 2025), GraphPFN extends these ideas to graphs by augmenting a tabular foundation model with attention-based message-passing adapters. This design allows GraphPFN to reuse strong feature modeling from tabular pretraining while capturing complex graph-specific patterns.

### 3.1 ARCHITECTURE

Our model architecture extends the tabular foundation model LimiX (Zhang et al., 2025) by adding attention-based message-passing layers to each of its transformer blocks. This strategy is inspired by

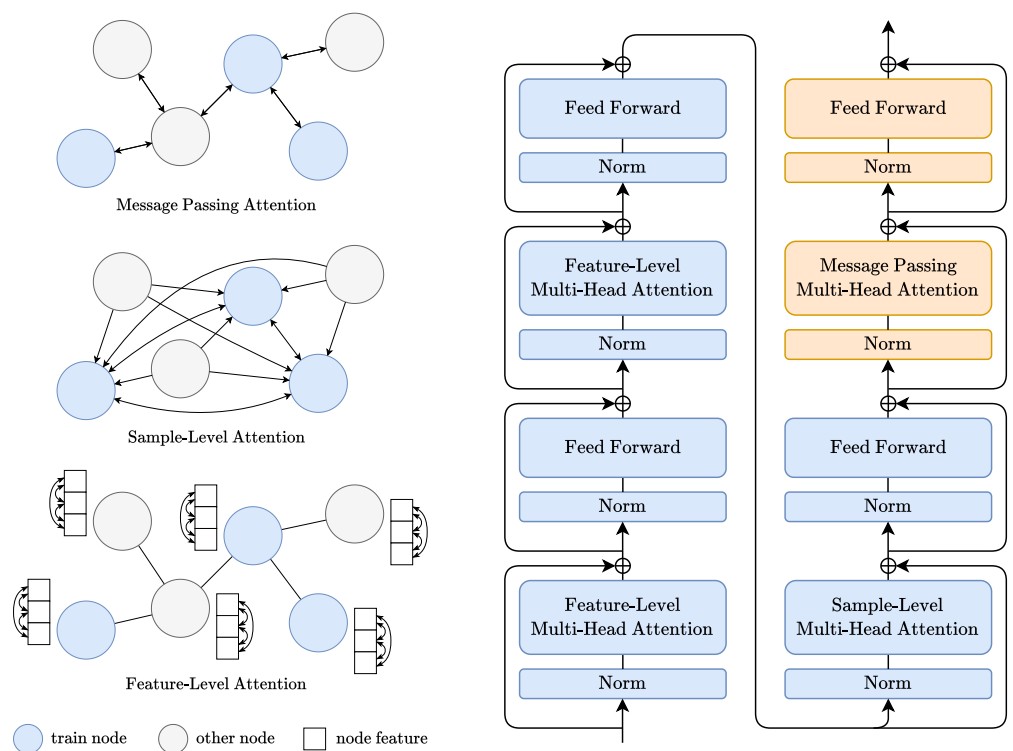

(a) Different attention mechanisms in GraphPFN.    (b) Base neural block in GraphPFN architecture.

Figure 1: An illustration of GraphPFN architecture.

recent findings that tabular foundation models can already learn patterns relevant for various graph tasks (Eremeev et al., 2025; Hayler et al., 2025). By initializing our model with a pretrained tabular foundation model instead of training from scratch, we leverage these learned representations, which significantly reduces computational costs while still achieving strong results.

Below we first summarize the LimiX backbone to clarify how GraphPFN represents samples (nodes) and features, and how attention flows in the base model. We then describe how the graph adapters modify this flow to leverage the graph topology.

**LimiX**    LimiX (Zhang et al., 2025) is a transformer-style foundation model for tabular data that departs from the common design of representing each sample with a single fixed-length embedding. Instead, it uses a multi-token representation: for every sample, each feature contributes one token,[2] yielding a token grid with one axis for features and one for samples. This design naturally handles a variable number of heterogeneous features in different datasets without changing and re-training the model.

A LimiX transformer block contains three attention layers, each followed by a element-wise feed-forward network (FFN). Two layers are feature-level multi-head attention (MHA) modules that operate within a sample, allowing all feature tokens of the same sample to attend to each other. And the third is a sample-level MHA that operates within a feature across samples, allowing tokens corresponding to the same feature to exchange information across the dataset. The feature-level MHAs enable rich interactions among features within each sample, while the sample-level MHA supports in-context learning by transferring information across samples for the same feature.

Attention masking at the sample level follows the standard PFN protocol: training (context) samples attend to all other training samples, and test (query) samples attend only to training samples. Thus, information can flow from train to test but not from test to train or between test samples. Feature-level attention within a sample is unmasked.

---

[2]In the current implementation, two features are grouped into one token, but we omit this detail for clarity.

**Graph adapters** To inject graph structure information without disrupting the LimiX's tokenization, we add a message-passing adapter that implements scaled dot product attention between neighboring nodes to the end of every LimiX transformer block (see Figure 1 for an illustration). Note that we use the scaled dot product attention that is identical to the original Transformer attention (Vaswani et al., 2017) except for being restricted to 1-hop graph neighborhoods. Intuitively, our message-passing adapter performs a second, graph-structure-aware round of sample-level attention: tokens may exchange information only along the observed edges. In contrast to the global sample-level attention of LimiX (which is masked by the PFN protocol), the graph adapter is masked by the adjacency and therefore routes information locally, from each node to its neighbors. Because we keep the per-feature token representation intact, the adapter runs over the sample axis for each feature token independently, using the same graph mask across features.

This design complements the global PFN-style attention by adding a local channel that is common in graph learning. We process the entire graph jointly and the graph adapter allows bidirectional exchange between labeled and unlabeled nodes along edges. Similar to the classic GNNs, these message-passing layers allow the model to capture complex graph dependencies that cannot be captured by hand-crafted features.

Each adapter is implemented as a sparse, multi-head attention module with the adjacency as its mask (two nodes attend if and only if an edge connects them). Similar to other attention modules, it is followed by a feed-forward network independently applied to each token. Both FFN and message-passing layer are wrapped with residual connections (He et al., 2016) and layer normalization (Ba et al., 2016), mirroring the structure of the LimiX blocks for stable optimization.

## 3.2 PRETRAINING

GraphPFN was pretrained following the PFN framework (Müller et al., 2021; Hollmann et al., 2023; 2025), using continuous pretraining from the LimiX checkpoint. Specifically, we generated 4,000,000 synthetic datasets according to the prior described below and used them for pretraining. The pretraining process was conducted for 500,000 steps on 8 NVIDIA A100 80G GPUs, with each GPU processing one synthetic dataset per step. The total pretraining time was approximately 6 days.

To monitor the progress of pretraining, we evaluated the in-context learning performance of Graph-PFN every 100 steps on several datasets from the GraphLand benchmark (Bazhenov et al., 2025) (see Section 5.1 for the evaluation procedure). We emphasize that these evaluations were used only for a post-hoc analysis of training progress: test set performance was not used for early stopping or any other form of model selection. The evaluation results can be found in Figure 2.

**Objective** We optimize a joint objective that combines the PFN supervised loss with the masked graph modeling (MGM) loss (Li et al., 2023). For the supervised PFN term, for each dataset we sample a random set of context nodes, compute predictions for all other nodes, and minimize the cross-entropy loss on them.

To encourage more graph-aware representations, inspired by the success of GNN self-supervised pretraining, we add the masked graph modeling term from Li et al. (2023). Specifically, we randomly sample a fraction $p = 0.1$ of edges as positive samples, remove these edges from the input graph, and uniformly sample an equal number of unconnected node pairs as negative examples. We then train the model with the cross-entropy loss to distinguish positive and negative edges. For this purpose, we add an additional MLP head on top of the target embeddings from the last layer. The total loss is the sum of the supervised loss and the MGM loss, with a coefficient of 0.1 applied to the MGM term.

**Details** To ensure training stability and retain the strong feature modeling capabilities of LimiX, we froze all model layers except for the graph adapters, which were the only components updated during pretraining. Optimization was performed using the Adam optimizer (Kingma & Ba, 2014) with a constant learning rate of $\gamma = 3 \cdot 10^{-4}$ and a linear warmup schedule over the first 10,000 steps.

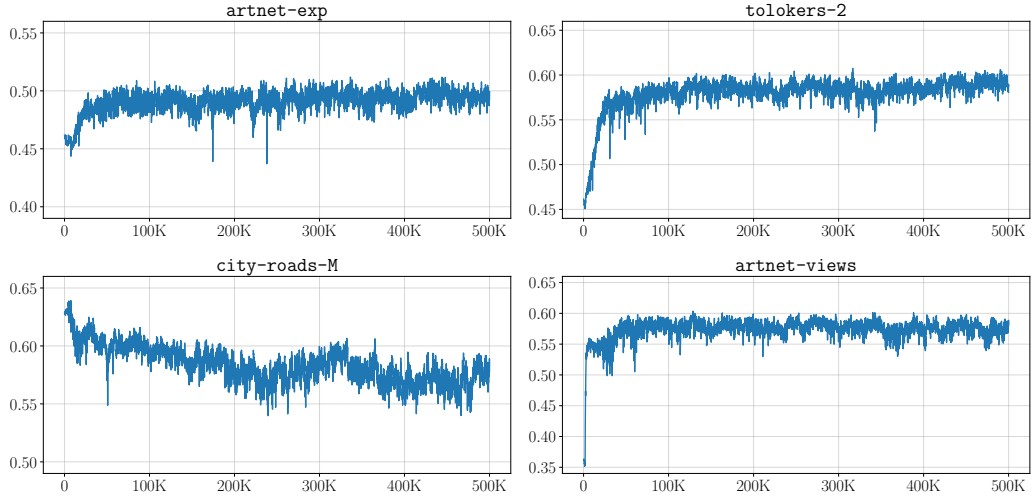

Figure 2: In-context learning performance of intermediate checkpoints of GraphPFN. The $x$-axis represents the number of steps, and the $y$-axis represents the metric on the test set.

## 4 GRAPH PRIOR

As discussed above, GraphPFN is based on the prior-data fitted networks (PFNs) framework (Müller et al., 2021). In this approach, the model is pretrained on a large number of synthetic graph datasets sampled from a chosen prior distribution. Because the pretrained model aims to approximate the posterior predictive distribution, it is crucial to design a high-quality and realistic prior. In the following sections, we describe the prior used for pretraining GraphPFN. First, we explain our method for generating realistic graph structures. Then, we describe how we use these graphs to generate node attributes and targets.

### 4.1 STRUCTURE GENERATION

Our main aim is to generate graph structures similar to real-world graphs. After examining graphs from a range of graph machine learning datasets, we find that most of them exhibit strong cluster (community) structure, which in general is a common feature of real-world graphs (Girvan & Newman, 2002). Thus, as the basis for our graph generation process, we use the degree-corrected stochastic block model (SBM) (Karrer & Newman, 2011) that can generate graphs with community structure. However, we find that graphs generated from the degree-corrected SBM exhibit too "clean"-looking and well-defined clusters that can be visualized as several well-separated "balls", while clusters in real-world graphs are often much more "rough"-looking, with more complex shapes and often overlapping with each other. Thus, to obtain graph structures similar to real-world ones, we design a novel method that combines multiple SBMs. First, we generate several *first-level* graphs from SBMs with different parameters. Then, we generate a *second-level* graph from another SBM, such that this second-level graph has the number of nodes equal to the sum of the numbers of nodes in all first-level graphs. We then randomly assign each node from the first-level graphs to a unique node in the second-level graph, thus essentially constructing a bijection $f$ between first-level and second-level graph nodes. Then, we transfer each edge from the first-level graphs to the second-level graph by creating a new edge in the second-level graph between the corresponding nodes, i.e., if there was an edge between nodes $u$ and $v$ in the first-level graphs, then we create an edge between the nodes $f(u)$ and $f(v)$ in the second-level graph. The obtained second-level graph with additional edges combines multiple graphs generated from different SBMs and exhibits clusters of nodes with complex shapes and overlaps that we were looking for.

Further, we observe that most graphs from graph benchmarks exhibit core-periphery structure, where the core is composed of multiple relatively dense clusters, but there are also many low-degree peripheral nodes. Such structure is known to be common for real-world networks (Zhang et al., 2015).

Table 1: The key statistics of the considered graph datasets.

| name | # nodes | # edges | # features | mean degree | task | # classes | homophily | feature type |
|---|---|---|---|---|---|---|---|---|
| tolokers-2 | 11,758 | 519,000 | 16 | 88.3 | cls. | 2 | no | tabular |
| artnet-exp | 50,405 | 280,348 | 75 | 11.1 | cls. | 2 | no | tabular |
| hm-prices | 46,563 | 10,730,995 | 41 | 460.9 | reg. | N/A | no | tabular |
| city-roads-M | 57,073 | 107,104 | 26 | 3.8 | reg. | N/A | yes | tabular |
| artnet-views | 50,405 | 280,348 | 50 | 11.1 | reg. | N/A | no | tabular |
| city-reviews | 148,801 | 1,165,415 | 37 | 15.7 | cls. | 2 | yes | tabular |
| avazu-ctr | 76,269 | 10,984,077 | 260 | 288.0 | reg. | N/A | no | tabular |
| twitch-views | 168,114 | 6,797,557 | 4 | 80.9 | reg. | N/A | no | tabular |
| facebook | 22,470 | 170,823 | 128 | 15.2 | cls. | 4 | yes | text-based |
| amazon-ratings | 24,492 | 93,050 | 300 | 7.6 | cls. | 5 | no | text-based |
| questions | 48,921 | 153,540 | 301 | 6.3 | cls. | 2 | no | text-based |
| wiki-cs | 11,701 | 215,603 | 300 | 36.9 | cls. | 10 | yes | text-based |
| pubmed | 19,717 | 44,324 | 500 | 4.5 | cls. | 3 | yes | text-based |

While our method of combining multiple graphs generated from SBM produces realistic node clusters, it produces a relatively small number of peripheral nodes. Thus, we augment the approach discussed above with a preferential attachment process (Price, 1965; 1976). Specifically, we run a modified Barabási–Albert (BA) process to add additional low-degree nodes to the graph (i.e., we use the graph obtained thus far as the initialization for the BA process). While the original BA process (Albert & Barabási, 2002) fixes the degree of newly-created nodes for the entire process, we treat this degree as a random variable and generate new nodes with different initial degrees.

Our method has many hyperparameters such as the number and size of blocks for SBMs or their degree sequences. Similar to prior works on PFNs (Hollmann et al., 2023; 2025; Qu et al., 2025), we define probability distributions for each hyperparameter and sample new hyperparameters for each synthetic graph, which allows us to generate a diverse graphs. At the same time, we can easily set bounds for sizes, densities, or maximum degrees of the generated graphs, allowing us to ensure that the graphs fit the desirable constraints.

Since we need many thousands of synthetic graphs for training, the efficiency of generation becomes a concern. We utilize the implementations of the degree-corrected SBM and the BA process from the `graph-tool` library (Peixoto, 2017), which are highly efficient and allow even a single CPU core to generate multiple graphs in a second according to our process.

We provide example visualizations of several graphs generated by our process in Appendix C.

## 4.2 Attribute generation

We generate features and targets for synthetic graphs with a neural structural causal model (SCM) that extends the MLP-based SCM of Qu et al. (2025); Hollmann et al. (2023). As a starting point, we follow the TabICL protocol: we sample an MLP architecture (number of layers, dimension of hidden layers, activation function) and its weights at random, draw random inputs, propagate them through the network, and then designate a random subset of neurons as observed features and another random neuron as target, leaving the rest as latent variables. This yields a broad family of causal mechanisms in which features and targets can depend on each other and on latent confounders. We refer to Qu et al. (2025); Hollmann et al. (2023) for further details.

To make attributes also depend on the graph structure, we extend this SCM in two complementary ways. First, we introduce a mixture of MLP and GNN neurons. For each dataset, we sample a mixing probability $p \in \{0.0, 0.1, \dots, 0.9, 1.0\}$. At every hidden layer, we compute both an MLP transformation and a GNN layer. Each neuron is then independently assigned to be MLP-type or GNN-type, with probabilities $1 - p$ and $p$, respectively, and its value is taken from the corresponding transformation. This mechanism controls how strongly the generated variables depend on the graph. Second, with a given probability, we optionally augment the random inputs with Laplacian positional encodings (LapPE) (Dwivedi et al., 2020; Belkin & Niyogi, 2001) to further integrate graph structure into the data generation process.

Together, the mixed MLP/GNN neurons and optional LapPE inject graph information into the SCM while remaining close to the tabular prior. When $p = 0$ and LapPE is not used, the procedure reduces

to the TabICL-style tabular SCM, while larger $p$ and the inclusion of LapPE increase the influence of graph structure on both features and targets.

## 5 EXPERIMENTS

### 5.1 EXPERIMENTAL SETUP

**Datasets** In terms of dataset selection, we closely follow the experimental setup of G2T-FM (Eremeev et al., 2025). We evaluate two collections of datasets: (i) real-world datasets from the recently proposed GraphLand benchmark (Bazhenov et al., 2025); and (ii) some of the classic graph datasets. Together, these datasets cover node classification and regression, come from diverse application domains, include both homophilous and non-homophilous graphs,[3] span a range of densities and other graph structural properties. Table 1 lists datasets used in our study and summarizes their statistics. For all datasets, we use 10%/10%/80% train/validation/test splits. Due to current limitations of TFMs, we restrict our study to small- and medium-scale datasets and exclude classification tasks with more than 10 classes.[4]

In our evaluation, we run all experiments 10 times and report the mean and standard deviation of the model performance. We report average precision for binary classification tasks, accuracy for multiclass classification tasks, and $R^2$ for regression tasks. For all metrics, higher is better.

**Methods** In addition to the proposed GraphPFN, we evaluate the following methods:

- **LightGBM** (Ke et al., 2017), a strong tabular baseline, augmented with neighborhood feature aggregation (NFA) (Bazhenov et al., 2025) to incorporate information about the graph structure.
- **Classic GNNs**: GCN (Kipf & Welling, 2017), GraphSAGE (Hamilton et al., 2017), GAT (Veličković et al., 2018), neighborhood-attention Graph Transformer (GT) (Shi et al., 2021). Following Platonov et al. (2023b), we augment these models with residual connections (He et al., 2016), layer normalization (Ba et al., 2016), and MLP blocks, which have been shown to substantially improve the performance of classic GNNs (Luo et al., 2024; 2025). We perform extensive hyperparameter tuning for these models.
- **G2T-FM** (Eremeev et al., 2025) with TabPFNv2 (Hollmann et al., 2025) and LimiX (Zhang et al., 2025) as the backbones. To the best of our knowledge, these are the strongest publicly available graph foundation models for node-level tasks with non-textual features.

We evaluate graph foundation models in both in-context learning (ICL) and full finetuning (FT) settings. We also optionally augment GraphPFN with Laplacian positional encodings (LapPE) (which are always used by G2T-FM). For further details, please refer to Appendix D.

### 5.2 EXPERIMENTAL RESULTS

Table 2 and Table 3 present the results of our experiments. Below, we summarize and discuss our key observations. Additional experimental results are provided in Appendix B.

> **Observation 1** *The in-context learning performance of GraphPFN is promising. In particular, GraphPFN outperforms both classic GNNs and other GFMs on some datasets.*

For example, on `artnet-exp` and `city-reviews`, GraphPFN (ICL) outperforms both baselines trained from scratch and G2T-FM applied in the ICL mode. Moreover, on `artnet-exp`, GraphPFN (ICL) even outperforms finetuned G2T-FM.

At the same time, performance on some datasets like `hm-prices` or `city-roads-M` is significantly worse than that of G2T-LimiX. We hypothesize that this is caused by a mismatch between

---

[3]A graph is called homophilous if its edges tend to connect nodes with similar labels, see Newman (2003); Platonov et al. (2023a); Mironov & Prokhorenkova (2024) for details.

[4]In principle, TFMs can handle more than 10 classes via schemes such as error-correcting output codes, as proposed in Hollmann et al. (2025). For simplicity, we focus on datasets with at most 10 classes that are natively supported by existing TFMs.

Table 2: Evaluation results on GraphLand datasets under the RL (Random Low) data split. OOM stands for out-of-memory error on NVIDIA A100 80G GPU. We report average precision for binary classification tasks and $R^2$ for regression tasks.

| | tolokers-2 | artnet-exp | hm-prices | city-roads-M | artnet-views | city-reviews | avazu-ctr | twitch-views |
|---|---|---|---|---|---|---|---|---|
| LightGBM-NFA | $56.34 \pm 0.06$ | $46.13 \pm 0.03$ | $70.84 \pm 0.04$ | $61.18 \pm 0.03$ | $56.10 \pm 0.02$ | $78.53 \pm 0.01$ | $31.71 \pm 0.01$ | $60.14 \pm 0.01$ |
| GCN | $56.27 \pm 0.29$ | $44.86 \pm 0.34$ | $68.02 \pm 0.40$ | $58.82 \pm 0.24$ | $56.03 \pm 0.24$ | $77.81 \pm 0.14$ | $32.00 \pm 0.15$ | $75.51 \pm 0.05$ |
| GraphSAGE | $54.43 \pm 0.32$ | $45.14 \pm 0.34$ | $70.00 \pm 0.70$ | $59.44 \pm 0.26$ | $49.32 \pm 0.86$ | $78.17 \pm 0.09$ | $31.44 \pm 0.15$ | $66.29 \pm 0.31$ |
| GAT | $57.41 \pm 0.80$ | $45.06 \pm 0.49$ | $72.07 \pm 1.16$ | $59.86 \pm 0.19$ | $53.60 \pm 0.23$ | $77.74 \pm 0.20$ | $32.63 \pm 0.16$ | $72.89 \pm 0.25$ |
| GT | $56.98 \pm 0.53$ | $46.41 \pm 0.68$ | $69.44 \pm 0.89$ | $59.55 \pm 0.27$ | $53.37 \pm 0.43$ | $77.34 \pm 0.20$ | $31.11 \pm 0.47$ | $72.13 \pm 0.13$ |
| G2T-TabPFNv2 (ICL) | $60.42 \pm 0.27$ | $45.84 \pm 0.03$ | $66.68 \pm 0.09$ | $60.47 \pm 0.04$ | $58.75 \pm 0.15$ | $77.46 \pm 0.10$ | $26.38 \pm 0.07$ | $70.00 \pm 0.06$ |
| G2T-LimiX (ICL) | $61.48 \pm 0.30$ | $48.43 \pm 0.18$ | $74.96 \pm 0.06$ | $64.53 \pm 0.07$ | $60.95 \pm 0.10$ | $77.72 \pm 0.54$ | $32.39 \pm 0.14$ | $71.08 \pm 0.07$ |
| G2T-TabPFNv2 (FT) | $57.65 \pm 1.92$ | $47.31 \pm 0.59$ | $71.05 \pm 0.91$ | $63.08 \pm 0.28$ | $60.29 \pm 0.13$ | $79.12 \pm 0.21$ | $28.52 \pm 0.43$ | $74.06 \pm 0.16$ |
| G2T-LimiX (FT) | $61.17 \pm 0.49$ | $49.88 \pm 0.13$ | $76.32 \pm 0.17$ | $65.87 \pm 0.10$ | $62.12 \pm 0.10$ | $80.13 \pm 0.05$ | $33.94 \pm 0.34$ | $73.16 \pm 0.40$ |
| GraphPFN (ICL) | $58.95 \pm 0.02$ | $49.97 \pm 0.01$ | $68.02 \pm 0.02$ | $58.93 \pm 0.02$ | $58.87 \pm 0.01$ | $79.52 \pm 0.01$ | $21.67 \pm 0.05$ | $59.95 \pm 0.08$ |
| GraphPFN + LapPE (ICL) | $61.18 \pm 0.11$ | $49.68 \pm 0.04$ | $68.56 \pm 0.10$ | $58.61 \pm 0.41$ | $60.31 \pm 0.09$ | $79.73 \pm 0.08$ | $23.33 \pm 0.47$ | $63.66 \pm 0.77$ |
| GraphPFN (FT) | $60.77 \pm 0.72$ | $53.08 \pm 0.23$ | $79.15 \pm 0.33$ | $65.65 \pm 0.20$ | $64.15 \pm 0.21$ | OOM | OOM | OOM |
| GraphPFN + LapPE (FT) | $61.83 \pm 0.28$ | $53.38 \pm 0.18$ | $78.58 \pm 0.37$ | $64.09 \pm 0.38$ | $64.39 \pm 0.13$ | OOM | OOM | OOM |

Table 3: Evaluation results on classic graph datasets under a random $10\%/10\%/80\%$ train/val/test split. OOM stands for out-of-memory error on NVIDIA A100 80G GPU. We report average precision for binary classification tasks and accuracy for multiclass classification tasks.

| | facebook | amazon-ratings | questions | wiki-cs | pubmed |
|---|---|---|---|---|---|
| GCN | $91.26 \pm 0.19$ | $41.43 \pm 0.46$ | $15.42 \pm 0.63$ | $81.74 \pm 0.20$ | $85.46 \pm 0.18$ |
| GraphSAGE | $91.12 \pm 0.21$ | $40.07 \pm 0.50$ | $16.55 \pm 0.61$ | $81.50 \pm 0.26$ | $86.04 \pm 0.26$ |
| GAT | $92.61 \pm 0.20$ | $40.67 \pm 0.53$ | $16.75 \pm 0.63$ | $82.25 \pm 0.26$ | $84.81 \pm 0.22$ |
| GT | $91.71 \pm 0.21$ | $41.56 \pm 0.38$ | $14.03 \pm 0.86$ | $82.54 \pm 0.20$ | $84.95 \pm 0.18$ |
| G2T-TabPFNv2 (ICL) | $90.56 \pm 0.12$ | $40.63 \pm 0.19$ | $16.49 \pm 0.16$ | $76.61 \pm 0.57$ | $88.80 \pm 0.25$ |
| G2T-LimiX (ICL) | $91.29 \pm 0.14$ | $44.10 \pm 0.16$ | $15.31 \pm 0.77$ | $79.99 \pm 0.28$ | $88.96 \pm 0.18$ |
| G2T-TabPFNv2 (FT) | $91.73 \pm 0.28$ | $44.71 \pm 0.32$ | $19.07 \pm 0.53$ | $79.70 \pm 0.31$ | $90.46 \pm 0.11$ |
| G2T-LimiX (FT) | $92.16 \pm 0.18$ | $45.67 \pm 0.35$ | $20.19 \pm 0.30$ | $82.24 \pm 0.31$ | $89.91 \pm 0.48$ |
| GraphPFN (ICL) | $90.23 \pm 0.03$ | $43.71 \pm 0.03$ | $12.30 \pm 0.05$ | $77.29 \pm 0.03$ | $89.76 \pm 0.02$ |
| GraphPFN + LapPE (ICL) | $91.45 \pm 0.16$ | $42.64 \pm 0.22$ | $9.62 \pm 0.16$ | $79.13 \pm 0.37$ | $89.41 \pm 0.09$ |
| GraphPFN (FT) | $93.06 \pm 0.09$ | $46.18 \pm 0.17$ | $20.50 \pm 0.83$ | $82.17 \pm 0.22$ | OOM |
| GraphPFN + LapPE (FT) | $92.97 \pm 0.19$ | $45.78 \pm 0.41$ | $18.20 \pm 3.51$ | $82.15 \pm 0.33$ | OOM |

our prior and the downstream datasets. For example, during pretraining, GraphPFN has seen only graphs with up to 194,425 edges, while hm-prices contains 10,730,995 edges. As another example, the city-roads-M dataset presents a traffic network, which has a significantly different topology from the graphs in our prior. However, we believe that poor performance of GraphPFN on these datasets is not a fundamental limitation of our approach, but rather a direction for future work. By extending the prior and making it more diverse, one can potentially improve the performance of GraphPFN on the datasets where it currently does not achieve the best results.

> **Observation 2** *On datasets with up to* 50,000 *nodes, finetuned GraphPFN achieves state-of-the-art results.*

In particular, on 7 out of 10 datasets where we were able to finetune GraphPFN, it achieved the best results across all considered methods, while having competitive performance on the remaining 3 datasets. Notably, on several datasets, such as artnet-exp, hm-prices, and artnet-views, finetuned GraphPFN outperforms the second-best method by more than two percentage points.

We hypothesize that such strong performance comes from the ability of GraphPFN to capture complex graph patterns via message passing, unlike G2T-FM, which is limited to hand-crafted graph-based features. However, we also note that the pretraining procedure plays a crucial role in this success. Replacing the pretrained graph adapters with randomly initialized ones and finetuning this model for a specific downstream dataset leads to significantly weaker results, as detailed in Appendix B.

## 6 CONCLUSION

In this work, we propose GraphPFN: a prior-data fitted graph foundation model for node-level tasks. Following the PFN framework, GraphPFN was pretrained on synthetic datasets drawn from our novel prior over attributed graphs to perform predictions in the in-context learning regime. Our experiments show promising results for GraphPFN. Even in the ICL regime, GraphPFN often out-

performs classic GNNs and prior ICL GFMs. After finetuning, GraphPFN consistently achieves state-of-the-art results, sometimes bringing substantial improvements over the second-best method. We believe our work shows that pretraining graph foundation models on synthetic datasets drawn from a carefully designed prior can be a promising direction for developing truly generalizable graph foundation models. Despite promising results, the current implementation of GraphPFN has several limitations, which we discuss in Appendix A.

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

## A  LIMITATIONS AND FUTURE WORK

- GraphPFN is difficult to scale to very large datasets since its current implementation requires processing the whole dataset at once. This leads to significant memory consumption, which prevents us from, for example, finetuning GraphPFN on some graphs from our benchmark. Developing more scalable graph foundation models can be a promising direction for future work. For example, one can utilize more memory-efficient TFMs (like TabICL (Qu et al., 2025)) as backbones or combine GraphPFN with sampling methods (Hamilton et al., 2017; Zeng et al., 2019; Zou et al., 2019; Chiang et al., 2019; Zeng et al., 2021; Balin & Çatalyürek, 2023).

- The proposed graph prior does not cover graphs from specific domains like traffic networks. We hypothesize that this can be a key reason for the degraded performance on some datasets. Extending the prior with more diverse graph random models (e.g., geometric graphs (Barthélemy, 2011)) can further improve the results of GraphPFN and make its performance more robust.

- Due to substantial computational resources required to pretrain GraphPFN, our work has limited ablation. Investigating the importance of incorporating self-supervised objectives like masked graph modeling into the GraphPFN objective or analyzing different components of the graph prior can bring new insights to the field.

- GraphPFN inherits some limitations from its tabular backbone LimiX. For example, GraphPFN does not natively support handling more than 10 classes in multiclass classification. This limitation can be alleviated with further development of TFMs or by using approaches such as error-correcting output codes, as proposed in Hollmann et al. (2025).

- Currently, GraphPFN is limited to node-level prediction tasks and cannot handle link prediction or graph-level tasks (e.g., graph classification or regression). While we have taken a first step in this direction by adding a masked graph modeling head to GraphPFN during pretraining, GraphPFN still heavily relies on the presence of node-level labels to make predictions. Therefore, we cannot directly apply GraphPFN to link prediction tasks. Extending GraphPFN to other prediction tasks can be a promising direction for future research.

Table 4: Additional comparison of finetuned GraphPFN with finetuned LimiX with randomly initialized graph adapters (GA).

| | tolokers-2 | artnet-exp | hm-prices | city-roads-M | artnet-views | facebook | amazon-ratings | questions | wiki-cs |
|---|---|---|---|---|---|---|---|---|---|
| GraphPFN (FT) | $60.77 \pm 0.72$ | $53.08 \pm 0.23$ | $79.15 \pm 0.33$ | $65.65 \pm 0.20$ | $64.15 \pm 0.21$ | $93.06 \pm 0.09$ | $46.18 \pm 0.17$ | $20.50 \pm 0.83$ | $82.17 \pm 0.22$ |
| LimiX + GA (FT) | $46.19 \pm 0.73$ | $46.79 \pm 0.10$ | $70.15 \pm 0.44$ | $63.22 \pm 0.48$ | $54.02 \pm 0.57$ | $90.82 \pm 0.39$ | $39.81 \pm 0.40$ | $15.88 \pm 4.00$ | $78.35 \pm 0.73$ |

## B    ADDITIONAL RESULTS

Since GraphPFN achieves strong performance in the finetuning regime, one may hypothesize that the performance comes solely from the powerful graph adapters, but not from the pretraining procedure. To test this hypothesis, we consider the following model. We start with LimiX and add graph adapters, so the architecture exactly matches that of GraphPFN. But instead of using pretrained weights for graph adapters, we employ random weights. In order to stabilize training, we initialized the last layers in all graph adapters with zeros, ensuring that the random initialization does not break the model. After that, we finetune this model following exactly the same protocol as GraphPFN. The results of this model and a comparison with GraphPFN are presented in Table 4. One can see that using random adapters instead of pretrained ones leads to significant drops in performance, supporting the importance of pretraining for achieving the strong performance of the finetuned GraphPFN.

## C    SYNTHETIC GRAPH EXAMPLES

We design our prior to generate graphs that are both realistic and diverse. In Figures 3 and 4, we provide examples of our synthetic graphs. Note that these graphs tend to exhibit both community structure and core-periphery structure. Our graph generation process allows us to easily control various graph properties such as their size or density.

## D    IMPLEMENTATION DETAILS

Since some datasets (specifically, `amazon-ratings` and `questions`) with text-embedding features have relatively high feature dimensionality, which prevented us from directly finetuning on these datasets, we applied PCA to reduce the feature dimensionality to 64. We did not apply PCA to the `pubmed` dataset, since in our preliminary experiments applying PCA to that dataset led to degraded performance.

## E    LLM USAGE

LLMs have been used for proofreading and minor stylistic editing of the paper; the authors are responsible for all the content.

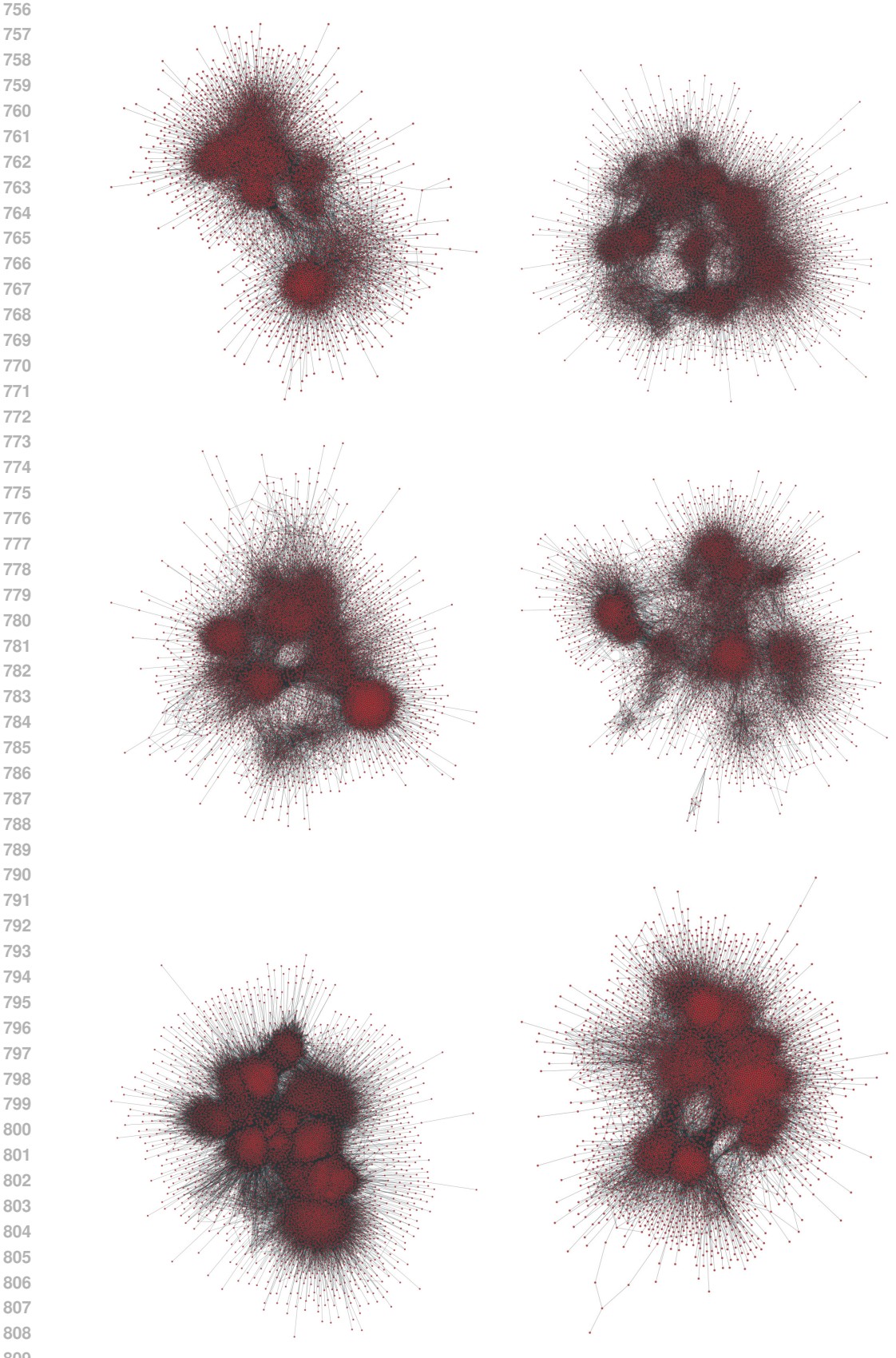

Figure 3: Example visualizations of denser graphs from our prior.

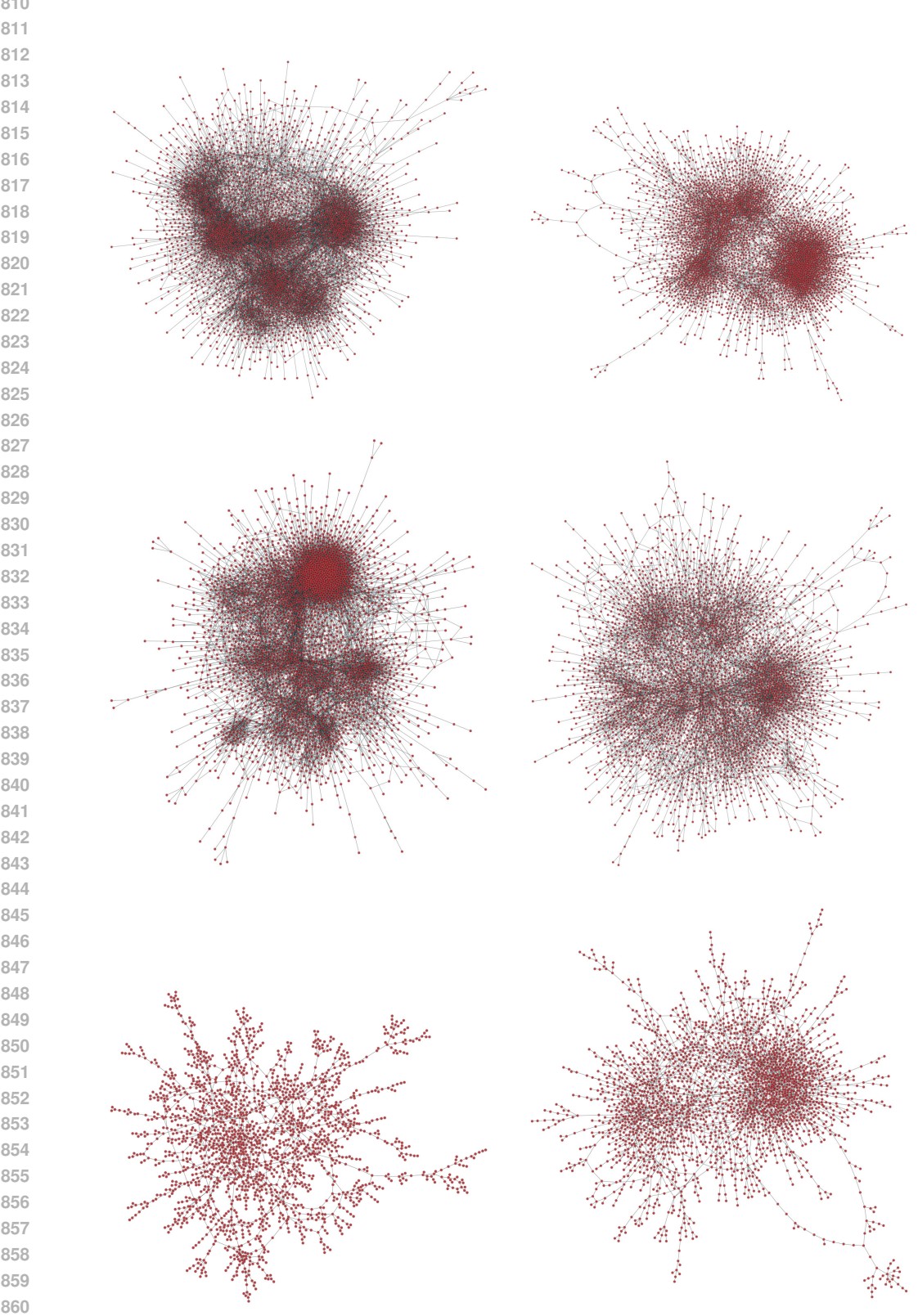

Figure 4: Example visualizations of sparser graphs from our prior.

