# OpenReview forum: "GraphPFN: A Prior-Data Fitted Graph Foundation Model"
_ICLR.cc/2026/Conference — Submitted to ICLR 2026_

### Official Review · Reviewer_4qnR · 2025-10-29

**Soundness:** 2
**Presentation:** 3
**Contribution:** 2
**Rating:** 2
**Confidence:** 4

**Summary:**

The paper proposes GraphPFN, which augments the tabular FM LimiX with attention-based message-passing adapters and pretrains them (PFN style) on synthetic graphs drawn from a novel prior that mixes degree-corrected SBMs, preferential attachment (to induce core-periphery), and graph-aware SCMs (MLP/GNN-mixed, optional LapPE). It shows promising ICL on some datasets and SOTA after finetuning on graphs up to ~50k nodes, though several settings hit OOM and TFMs’ class-count limits remain.

**Strengths:**

1. The paper proposes a clear architectural idea: lightweight graph adapters (masked attention over adjacency) added to each LimiX block; retains PFN ICL while enabling learned structural reasoning.
2. The paper combines multi-SBM with preferential attachment to mimic community + core-periphery; attributes from graph-aware SCM (MLP/GNN mix, optional LapPE).
3. The pretraining scale is large with 4M synthetic samples and 500k steps.

**Weaknesses:**

1. The efficiency is still the bottleneck. The model processes the whole graph jointly; OOM prevents FT on multiple datasets and the paper itself flags poor scaling. Provide rigorous memory/throughput curves and practical remedies.
2. Task breadth is narrow. Despite an MGM head, the method remains node-only; no on-main link/graph-level results, so the “foundation” scope is limited.
3. The baselines are limited. Not enough recent graph transformers or foundation models are compared. This could make the sota claim less effective.
4. ICL trails G2T-LimiX on heavy, structurally different graphs (e.g., hm-prices, city-roads-M), and authors attribute this to prior mismatch—but evidence and how to solve the mismatch is lack of discussion.

**Questions:**

1. Can the pretrained model be extended to other graph-related tasks? And what kind of changes are needed for the pretraining datasets and methods.
2. Can you integrate sampling (random walk) or chunked/sparse attention to avoid whole-graph processing?
3. What is performance vs. number/selection of labeled context nodes and class imbalance?
4. Could you please ablate MGM weight, LapPE during pretraining vs. downstream, and adapter placement per block; quantify where LapPE helps/hurts?
5. Could the author compare the method with more up-to-date baselines?

---

> ### Author Response · Authors · 2025-11-21
> **Official Comment (Part 1/2)**
>
> Thank you for your review. We address your questions and concerns below. Please also see the general response where we elaborate on why we believe GraphPFN is an important step for GFMs.
>
> > **W1**. The efficiency is still the bottleneck. The model processes the whole graph jointly; OOM prevents FT on multiple datasets and the paper itself flags poor scaling. Provide rigorous memory/throughput curves and practical remedies.
>
> We acknowledge that, in its current form, GraphPFN has issues with scalability and efficiency. Yet, as we write in our general response, we believe that now it is more important to demonstrate the applicability of this approach in principle, and we leave scaling to larger graphs to future research.
>
> > **W2**. Task breadth is narrow. Despite an MGM head, the method remains node-only; no on-main link/graph-level results, so the “foundation” scope is limited.
>
> > **Q1**. Can the pretrained model be extended to other graph-related tasks? And what kind of changes are needed for the pretraining datasets and methods.
>
> Currently, GraphPFN is designed to work only with node-level tasks, and we do not see an elegant way to extend it to link/graph-level tasks. Let us note that some previous works like TS-GNN, GCOPE and G2T-FM are also suitable only for node-level tasks and yet they position themselves as graph foundation models. As shown in G2T-FM, many previously developed GFMs struggle to generalize to arbitrary node-level tasks from different domains. Thus, this task is already challenging enough. According to our evaluation, GraphPFN can successfully work with various graphs having different heterogeneous node features and under realistic, non-fewshot splits. We also refer to our general response where we elaborate on why we believe GraphPFN is an important step for GFMs.
>
> > **W3**. The baselines are limited. Not enough recent graph transformers or foundation models are compared. This could make the sota claim less effective.
>
> > **Q5**. Could the author compare the method with more up-to-date baselines?
>
> Regarding the comparison with other GFMs, please note that G2T-FM is actually one of the most up-to-date models, having been released just at the end of this summer. At the same time, as it is shown in the corresponding study, many other known GFMs, including OpenGraph, AnyGraph, GCOPE and TS-GNN, are significantly outperformed by G2T-FM and task-specific GNNs in the same setting that we use in our work. Further, it has been shown recently [1] that well-tuned classic GNNs can serve as strong baselines and in particular outperform global graph transformers. Thus, we believe that our comparison with G2T-FM and standard well-tuned GNNs is the most relevant and meaningful in terms of predictive performance. However, if you have any specific examples of the baselines worth considering in our empirical study, please let us know.
>
> > **W4**. ICL trails G2T-LimiX on heavy, structurally different graphs (e.g., hm-prices, city-roads-M), and authors attribute this to prior mismatch—but evidence and how to solve the mismatch is lack of discussion.
>
> Thank you for this observation! We have obtained preliminary results that pretraining GraphPFN on synthetic graphs of large size and higher density significantly improves its ICL performance on certain datasets. Please refer to the additional experimental results in our general comment. We believe these results support our hypothesis about particular drops in the ICL performance on some datasets due to the lack of large or dense enough graphs in the prior distribution. Thus, introducing such graphs at the pretraining stage of GraphPFN can be treated as one possible way to solve these problems.
>
> > **Q2**. Can you integrate sampling (random walk) or chunked/sparse attention to avoid whole-graph processing?
>
> Thank you for this comment! Since GraphPFN works in the ICL mode, it needs to store the whole training set in the context. Thus, when using $10\%/10\%/10\%$ train/val/test splits as in this work, we need to process at least $10\%$ of the graph, and sampling nodes from their neighborhood can make the situation much worse. So, although sampling is most likely one of the needed components for scaling, we suppose that it may be not enough to completely solve the problem of efficient inference.

---

> > ### Author Response · Authors · 2025-11-21
> > **Official Comment (Part 2/2)**
> >
> > > **Q3**. What is performance vs. number/selection of labeled context nodes and class imbalance?
> >
> > Could you please clarify your question so that we can address it properly?
> >
> > > **Q4**. Could you please ablate MGM weight, LapPE during pretraining vs. downstream, and adapter placement per block; quantify where LapPE helps/hurts?
> >
> > Thank you for your suggestion.
> >
> > Please note that an extensive ablation of each component can be costly. So, as for now, we provide ablations that we consider to be most important. Specifically, we provide ablation of the MGM component and presence of structural features (i.e., LapPE, node degree and PageRank) in the graph prior (StrucFeat). For each model, we also evaluate it in two setups: with and without LapPE as features. The results of these ablations are given in the table below. One can see that removing either MGM or StrucFeat leads to a performance drop on average. Regarding LapPE during evaluation, one can see that it significantly improves performance on certain datasets like `twitch-views`.
> >
> > |                                    | pubmed       | facebook     | amazon-ratings | questions    | wiki-cs      | tolokers-2   | city-reviews | artnet-exp   | hm-prices    | avazu-ctr    | city-roads-M | twitch-views | artnet-views | AR   |
> > | :--------------------------------- | :----------- | :----------- | :------------- | :----------- | :----------- | :----------- | :----------- | :----------- | :----------- | :----------- | :----------- | :----------- | :----------- | :--- |
> > | GraphPFN-1.1                       | 89.53 ± 0.03 | 91.21 ± 0.01 | 43.88 ± 0.03   | 18.67 ± 0.04 | 79.48 ± 0.05 | 60.01 ± 0.03 | 79.31 ± 0.01 | 50.30 ± 0.01 | 71.45 ± 0.02 | 30.07 ± 0.01 | 61.86 ± 0.03 | 62.37 ± 0.04 | 61.70 ± 0.01 | 3.38 |
> > | GraphPFN-1.1 w/o MGM          | 88.66 ± 0.03 | 89.92 ± 0.03 | 44.41 ± 0.03   | 19.65 ± 0.04 | 78.82 ± 0.03 | 59.44 ± 0.02 | 80.02 ± 0.01 | 50.85 ± 0.02 | 71.84 ± 0.02 | 29.59 ± 0.02 | 60.75 ± 0.13 | 59.30 ± 0.04 | 60.80 ± 0.01 | 4.08 |
> > | GraphPFN-1.1 w/o StrucFeat         | 89.52 ± 0.02 | 91.07 ± 0.02 | 43.75 ± 0.03   | 16.37 ± 0.06 | 78.55 ± 0.09 | 59.76 ± 0.06 | 79.06 ± 0.01 | 50.69 ± 0.02 | 70.79 ± 0.02 | 30.76 ± 0.01 | 61.87 ± 0.06 | 62.16 ± 0.04 | 61.57 ± 0.01 | 4.08 |
> > | GraphPFN-1.1 + LapPE               | 89.40 ± 0.15 | 91.00 ± 0.23 | 43.32 ± 0.15   | 12.48 ± 0.37 | 79.82 ± 0.25 | 61.47 ± 0.17 | 79.56 ± 0.06 | 50.26 ± 0.05 | 71.57 ± 0.06 | 30.83 ± 0.22 | 61.80 ± 0.15 | 69.08 ± 0.34 | 62.93 ± 0.04 | 2.92 |
> > | GraphPFN-1.1 + LapPE w/o MGM  | 89.12 ± 0.10 | 91.59 ± 0.12 | 44.72 ± 0.12   | 8.14 ± 0.37  | 79.61 ± 0.33 | 60.87 ± 0.21 | 79.97 ± 0.07 | 50.24 ± 0.09 | 71.84 ± 0.07 | 30.56 ± 0.21 | 61.67 ± 0.16 | 68.04 ± 0.37 | 62.17 ± 0.08 | 3.15 |
> > | GraphPFN-1.1 + LapPE w/o StrucFeat | 89.35 ± 0.15 | 91.49 ± 0.08 | 43.03 ± 0.20   | 11.23 ± 0.31 | 79.62 ± 0.13 | 60.16 ± 0.25 | 79.27 ± 0.08 | 50.76 ± 0.07 | 70.93 ± 0.11 | 31.70 ± 0.04 | 61.52 ± 0.13 | 69.57 ± 0.29 | 62.66 ± 0.08 | 3.31 |
> >
> > ***
> >
> > [1] Classic GNNs are Strong Baselines: Reassessing GNNs for Node Classification, NeurIPS 2024.

---

> > > ### Author Response · Authors · 2025-11-28
> > >
> > > Dear Reviewer 4qnR,
> > >
> > > We have recently finished benchmarking time and memory efficiency, please see the results in [this comment](https://openreview.net/forum?id=BLJ5DsJ0i6&noteId=g5ZRzaYx2x). If you have any other concerns remaining, please let us know, we would be happy to discuss them further.
> > >
> > > Best regards,
> > >
> > > Authors

---

### Official Review · Reviewer_X1vi · 2025-10-30

**Soundness:** 2
**Presentation:** 2
**Contribution:** 2
**Rating:** 2
**Confidence:** 4

**Summary:**

This paper proposes GraphPFN, a graph foundation model for node-level prediction that extends the tabular foundation model LimiX by adding attention-based message-passing adapters to each transformer block. The model is pretrained on 4 million synthetic graphs sampled from a prior that combines multiple stochastic block models with preferential attachment for structure generation, and augments tabular structured causal models with message-passing mechanisms for attribute generation.

**Strengths:**

Motivation:  The proposed method combines multiple degree-corrected SBMs to address the "too clean" cluster problem in single-SBM generations, and the preferential attachment augmentation naturally to create the core-periphery structure in social and information networks, which sounds interesting.

Method: Rather than training a graph model from scratch, the authors initialize from LimiX and add sparse message-passing adapters that respect the per-feature tokenization. The motivation seems reasonable since it lets the model learn what to aggregate rather than relying entirely on hand-crafted features like G2T-FM does.

Empirical results: The overall performance is not bad if fully fine tuned.

**Weaknesses:**

Scalability: The scalability issues are frankly concerning for something claiming to be a foundation model. GraphPFN hits out-of-memory errors on 3 of 11 datasets during finetuning, that's failing on 27% of your own benchmark. What's worse is that these failures happen at 148K-168K nodes, I don't think these graph datasets are big enough comparing with OGB datasets. While the authors acknowledge this in Appendix A, but this isn't just a limitation to note. It fundamentally undermines the claim of "foundation model"  which should be scalable. The paper needs a more concrete solution (even if preliminary).

Prior analysis: The prior coverage problem reveals a deeper issue with the core contribution. When GraphPFN performs dramatically worse than G2T-LimiX on datasets like hm-prices or city-roads-M, and the explanation is "The proposed graph prior does not cover graphs from specific domains". That's a problem with the prior design, not just an implementation detail. The manuscript provides no quantitative validation that synthetic graphs match real-world statistics like degree distribution, clustering coefficient, or assortativity. The prior appears tailored to social/information networks without demonstrated generalization to physical/spatial networks. The core claim that pretraining on synthetic graphs from a well-designed prior is effective lacks sufficient support when systematic failures occur on specific graph types.

Insufficient ablation of critical design choices: The manuscript lacks ablations for several key decisions. The rationale for adding adapters to every LimiX block rather than fewer layers is unexplored. The choice to freeze LimiX layers during pretraining instead of joint training lacks justification through empirical comparison. The individual contribution of the masked graph modeling objective versus supervised PFN loss remains unclear, and the 0.1 coefficient for MGM appears arbitrary without sensitivity analysis. Performance sensitivity to the number of pretraining graphs (4M) and steps (500K) is unexplored despite Figure 2 showing continued improvement at 500K steps. The effect of prior hyperparameters like mixing probability p or LapPE inclusion frequency lacks systematic investigation. Furthermore, the compared GNN models are just so limited and many current state-of-the-art models are not included especially graph transformers and heterophilic GNNs.

Model comparison: G2T-FM uses Laplacian positional encodings (LapPE) in all experiments while GraphPFN do not use them at all, creating uneven comparison where G2T-FM accesses global positional information unavailable to GraphPFN base variant. G2T-FM employs neighborhood-aggregated features computed via explicit aggregation while GraphPFN relies on learned message passing. These differences prevent isolating whether improvements stem from trainable message passing specifically versus superior feature engineering. Further ablation study on this aspect should be conducted.

Weak in-context learning performance without systematic analysis: GraphPFN's ICL performance is substantially worse than G2T-LimiX on multiple datasets, sometimes by large margins (e.g., hm-prices: 68.02 vs. 74.96). This indicates the model has not fully learned to perform ICL on graphs, this really undermines a key promise of the PFN framework. If ICL performance is unreliable, GraphPFN's value as a zero-shot foundation model is limited, reducing it primarily to a pretrained initialization for finetuning. The manuscript provides no systematic analysis correlating ICL success with graph properties like size, density, or community structure to understand when the approach is applicable.

Validation of graph prior quality: The manuscript lacks quantitative validation that synthetic graphs match real-world distributions. Figures 3-4 provide only qualitative visualizations without comparing metrics like degree distribution, clustering coefficient, or graph diameter against real datasets. An analysis demonstrating that real datasets fall within the statistical distribution of synthetic graphs would substantiate claims about prior quality and coverage. The absence of such validation makes it difficult to assess whether poor performance on certain datasets stems from fundamental approach limitations or simply inadequate prior design.

Presentation: The overall presentation has many problems. 1) Captions are sloppy and not informative. 2) Some terms are not defined such as "OOM" even though I understand what is stands for but it should be clearly defined.

Overall, I feel that this paper is only half-finished, as many important aspects, especially those related to scalability, are not thoroughly addressed. More importantly, while the manuscript claims to focus on graph foundation models, it only evaluates node-level tasks, which makes its scope far too narrow.

**Questions:**

See weakness

---

> ### Author Response · Authors · 2025-11-21
> **Official Comment (Part 1/4)**
>
> Thank you for your review. We address your questions and concerns below. Please also see the general response where we elaborate on why we believe GraphPFN is an important step for GFMs.
>
> > **W1**. Scalability: The scalability issues are frankly concerning for something claiming to be a foundation model. GraphPFN hits out-of-memory errors on 3 of 11 datasets during finetuning, that's failing on 27% of your own benchmark. What's worse is that these failures happen at 148K-168K nodes, I don't think these graph datasets are big enough comparing with OGB datasets. While the authors acknowledge this in Appendix A, but this isn't just a limitation to note. It fundamentally undermines the claim of "foundation model" which should be scalable. The paper needs a more concrete solution (even if preliminary).
>
> We acknowledge that, in its current form, GraphPFN has issues with scalability and efficiency. Yet, as noted in our general response, we believe that now it is more important to demonstrate the applicability of this approach in principle, so we leave scaling to larger graphs for future research.
>
> > **W2**. Prior analysis: The prior coverage problem reveals a deeper issue with the core contribution. When GraphPFN performs dramatically worse than G2T-LimiX on datasets like hm-prices or city-roads-M, and the explanation is "The proposed graph prior does not cover graphs from specific domains". That's a problem with the prior design, not just an implementation detail. The manuscript provides no quantitative validation that synthetic graphs match real-world statistics like degree distribution, clustering coefficient, or assortativity. The prior appears tailored to social/information networks without demonstrated generalization to physical/spatial networks. The core claim that pretraining on synthetic graphs from a well-designed prior is effective lacks sufficient support when systematic failures occur on specific graph types.
>
> We acknowledge that our prior is tailored to social/information networks, so it is reasonable to expect that GraphPFN would not work well on spatial networks like `city-roads-M`. However, this does not affect our claim that “pretraining on synthetic graphs from a well-designed prior is effective”, since the designed prior appeared to be suitable for most other domains. We also conducted additional experiments finetuning the pretrained model on synthetic datasets with larger number of nodes and higher average density. Our results show that this procedure significantly improves the predictive performance on certain datasets. While in some cases the resulting model still does not match the performance of G2T-LimiX (ICL), we believe that these results support our hypothesis that poor performance on certain datasets were caused by mismatch in the number of nodes and average degree between our prior and these particular datasets. For further details, please refer to the general comment with additional experiments.

---

> ### Author Response · Authors · 2025-11-21
> **Official Comment (Part 2/4)**
>
> > **W6**. Validation of graph prior quality: The manuscript lacks quantitative validation that synthetic graphs match real-world distributions. Figures 3-4 provide only qualitative visualizations without comparing metrics like degree distribution, clustering coefficient, or graph diameter against real datasets. An analysis demonstrating that real datasets fall within the statistical distribution of synthetic graphs would substantiate claims about prior quality and coverage. The absence of such validation makes it difficult to assess whether poor performance on certain datasets stems from fundamental approach limitations or simply inadequate prior design.
>
> Thank you for this insightful suggestion!
>
> We believe that a well-designed graph prior is one of the key components for building GFMs, and its analysis should be of great interest. Below, we make first steps in this direction and compare several properties for real-world and synthetic datasets.
>
> Specifically, we analyze following properties:
> - `n_nodes`: number of nodes in the graph
> - `n_edges`: number of edges in the graph
> - `n_features`: number of node features
> - `avg_degree`: average degree of nodes in the graph
> - `target_homophily`: unbiased homophily of labels (for classification datasets)
> - `target_assortativity`: Pearson correlation coefficient for targets (for regression datasets)
> - `degree_powerlaw_r2`: R^2 of linear model fitted for cumulative degree distribution in log-log coordinates
> - `clustering_coefficient`: global clustering coefficient
> - `avg_pairwise_distance`: estimate of average pairwise distances between nodes
>
> We provide the values of the above properties for real-world datasets, as well as statistics of the corresponding distributions for synthetic datasets.
>
> |                        | tolokers-2 | city-reviews | artnet-exp | hm-prices | avazu-ctr | city-roads-M | twitch-views | artnet-views |
> | :--------------------- | :--------- | :----------- | :--------- | :-------- | :-------- | :----------- | :----------- | :----------- |
> | n_nodes                | 11758      | 148801       | 50405      | 46563     | 76269     | 57073        | 168114       | 50405        |
> | n_edges                | 519000     | 1165415      | 280348     | 10730995  | 10984077  | 107104       | 6797557      | 280348       |
> | n_features             | 16         | 37           | 75         | 41        | 260       | 24           | 4            | 50           |
> | avg_degree             | 88.28      | 15.66        | 11.12      | 460.92    | 288.04    | 3.75         | 80.87        | 11.12        |
> | target_homophily       | 0.10       | 0.69         | 0.28       | nan       | nan       | nan          | nan          | nan          |
> | target_assortativity   | nan        | nan          | nan        | 0.12      | 0.18      | 0.74         | -0.41        | 0.19         |
> | degree_powerlaw_r2     | 0.83       | 0.92         | 0.70       | 0.84      | 0.90      | 0.52         | 0.97         | 0.70         |
> | clustering_coefficient | 0.23       | 0.26         | 0.03       | 0.27      | 0.24      | 0.00         | 0.02         | 0.03         |
> | avg_pairwise_distance  | 2.84       | 4.78         | 4.34       | 2.39      | 3.72      | 129.89       | 2.84         | 4.43         |
>
>
> |                        | mean     | std      | min   | Q25     | median   | Q75      | max       |
> | :--------------------- | :------- | :------- | :---- | :------ | :------- | :------- | :-------- |
> | n_nodes                | 2784.49  | 2391.41  | 42.00 | 893.00  | 1919.00  | 4042.25  | 9971.00   |
> | n_edges                | 45570.56 | 89759.77 | 43.00 | 3273.00 | 11346.00 | 41739.25 | 926724.00 |
> | n_features             | 52.30    | 27.67    | 4.00  | 28.00   | 52.00    | 76.00    | 100.00    |
> | avg_degree             | 25.85    | 32.01    | 2.05  | 3.67    | 11.18    | 36.69    | 207.58    |
> | target_homophily       | 0.03     | 0.23     | -1.00 | -0.01   | 0.01     | 0.09     | 0.91      |
> | target_assortativity   | 0.06     | 0.17     | -0.59 | -0.01   | 0.01     | 0.09     | 0.87      |
> | degree_powerlaw_r2     | 0.82     | 0.14     | 0.41  | 0.71    | 0.85     | 0.94     | 1.00      |
> | clustering_coefficient | 0.12     | 0.12     | 0.00  | 0.03    | 0.08     | 0.19     | 0.62      |
> | avg_pairwise_distance  | 4.26     | 1.86     | 1.83  | 2.78    | 3.64     | 5.42     | 17.44     |
>
> As can be seen, for certain characteristics (like `degree_powerlaw_r2` or `avg_pairwise_distance`), our prior mostly covers the values encountered in real-world datasets. At the same time, several other properties of the prior like homophily or average degree, tend to be less aligned with those of real graphs. We believe such insights may allow for designing prior better suited for specific datasets.

---

> > ### Author Response · Authors · 2025-11-21
> > **Official Comment (Part 3/4)**
> >
> > > **W3**. ​​Insufficient ablation of critical design choices: The manuscript lacks ablations for several key decisions.
> >
> > > The rationale for adding adapters to every LimiX block rather than fewer layers is unexplored. The choice to freeze LimiX layers during pretraining instead of joint training lacks justification through empirical comparison.
> >
> > Thank you for your thoughtful suggestions. Due to the high computational cost of pretraining GraphPFN, it is hard to ablate all possible design choices. We decided to incorporate adapters to all TFM blocks since this is the most flexible solution: in theory, the model can learn zero weights for unnecessary adapters.
> > Next, we believe that, during the first steps of pretraining, the TFM backbone should be frozen, since it already brings various strong capabilities, such as processing diverse heterogeneous node features and making predictions via in-context learning, so we want to preserve them as much as possible. However, we also believe that it can be unfrozen later to improve the model adaptation to graph-specific tasks. It is reasonable to conduct such an experiment, but due to the high computational cost, we cannot promise that we will manage to do this during the discussion period.
> >
> > > The individual contribution of the masked graph modeling objective versus supervised PFN loss remains unclear, and the 0.1 coefficient for MGM appears arbitrary without sensitivity analysis.
> >
> > Thank you for an insightful suggestion!
> >
> > Below, we provide an ablation by removing the MGM component. As one can see, this negatively affects the average performance, justifying the usage of MGM.
> >
> > |                              | pubmed       | facebook     | amazon-ratings | questions    | wiki-cs      | tolokers-2   | city-reviews | artnet-exp   | hm-prices    | avazu-ctr    | city-roads-M | twitch-views | artnet-views | AR   |
> > | :--------------------------- | :----------- | :----------- | :------------- | :----------- | :----------- | :----------- | :----------- | :----------- | :----------- | :----------- | :----------- | :----------- | :----------- | :--- |
> > | GraphPFN-1.1                 | 89.53 ± 0.03 | 91.21 ± 0.01 | 43.88 ± 0.03   | 18.67 ± 0.04 | 79.48 ± 0.05 | 60.01 ± 0.03 | 79.31 ± 0.01 | 50.30 ± 0.01 | 71.45 ± 0.02 | 30.07 ± 0.01 | 61.86 ± 0.03 | 62.37 ± 0.04 | 61.70 ± 0.01 | 2.62 |
> > | GraphPFN-1.1 w/o MGM         | 88.66 ± 0.03 | 89.92 ± 0.03 | 44.41 ± 0.03   | 19.65 ± 0.04 | 78.82 ± 0.03 | 59.44 ± 0.02 | 80.02 ± 0.01 | 50.85 ± 0.02 | 71.84 ± 0.02 | 29.59 ± 0.02 | 60.75 ± 0.13 | 59.30 ± 0.04 | 60.80 ± 0.01 | 2.92 |
> > | GraphPFN-1.1 + LapPE         | 89.40 ± 0.15 | 91.00 ± 0.23 | 43.32 ± 0.15   | 12.48 ± 0.37 | 79.82 ± 0.25 | 61.47 ± 0.17 | 79.56 ± 0.06 | 50.26 ± 0.05 | 71.57 ± 0.06 | 30.83 ± 0.22 | 61.80 ± 0.15 | 69.08 ± 0.34 | 62.93 ± 0.04 | 2.15 |
> > | GraphPFN-1.1 + LapPE w/o MGM | 89.12 ± 0.10 | 91.59 ± 0.12 | 44.72 ± 0.12   | 8.14 ± 0.37  | 79.61 ± 0.33 | 60.87 ± 0.21 | 79.97 ± 0.07 | 50.24 ± 0.09 | 71.84 ± 0.07 | 30.56 ± 0.21 | 61.67 ± 0.16 | 68.04 ± 0.37 | 62.17 ± 0.08 | 2.23 |
> >
> > > Performance sensitivity to the number of pretraining graphs (4M) and steps (500K) is unexplored despite Figure 2 showing continued improvement at 500K steps.
> >
> > Please note that detailed ablation can be costly. As for now, the learning curves provided in our paper, although they are reported only for the test subsets, can show an estimated effect of the amounts of pretraining data on the downstream performance, since the longer GraphPFN model pretrains, the more distinct synthetic datasets it observes.
> >
> > > The effect of prior hyperparameters like mixing probability p or LapPE inclusion frequency lacks systematic investigation.
> >
> > Please note that we sampled these parameters individually for each dataset (see lines 349-355). This is done in order to increase diversity of data used for pretraining which is generally assumed to be helpful. If our clarification does not address your concern, could you please clarify what type of analysis you would find helpful?
> >
> > > Furthermore, the compared GNN models are just so limited and many current state-of-the-art models are not included especially graph transformers and heterophilic GNNs.
> >
> > It has been shown in several recent papers that well-tuned classic GNN models provided with common architectural enhancements like normalization and residual connections usually outperform more sophisticated models. For instance, [1] shows that such classic models outperform global graph transformers. As shown in [2], classic GNNs with these enhancements outperform heterophily-specific models. In our work, we use these enhancements and hyperparameter tuning, so we believe that we use strong and up-to-date baselines.

---

> > > ### Author Response · Authors · 2025-11-21
> > > **Official Comment (Part 4/4)**
> > >
> > > > **W4**. Model comparison: G2T-FM uses Laplacian positional encodings (LapPE) in all experiments while GraphPFN do not use them at all, creating uneven comparison where G2T-FM accesses global positional information unavailable to GraphPFN base variant. G2T-FM employs neighborhood-aggregated features computed via explicit aggregation while GraphPFN relies on learned message passing. These differences prevent isolating whether improvements stem from trainable message passing specifically versus superior feature engineering. Further ablation study on this aspect should be conducted.
> > >
> > > Could you please clarify which ablation experiment you would find informative? Please note that one disadvantage of G2T-FM is that it heavily relies on hand-crafted features and we wanted to avoid that in GraphPFN. That is why we do not use NFA or other hand-crafted features and instead rely on learnable message passing adapters.
> > >
> > > > **W5**. Weak in-context learning performance without systematic analysis: GraphPFN's ICL performance is substantially worse than G2T-LimiX on multiple datasets, sometimes by large margins (e.g., hm-prices: 68.02 vs. 74.96). This indicates the model has not fully learned to perform ICL on graphs, this really undermines a key promise of the PFN framework. If ICL performance is unreliable, GraphPFN's value as a zero-shot foundation model is limited, reducing it primarily to a pretrained initialization for finetuning. The manuscript provides no systematic analysis correlating ICL success with graph properties like size, density, or community structure to understand when the approach is applicable.
> > >
> > > First, please see our general comment in which we share additional experiments with GraphPFN pretraining: with some improvements of the learning procedure we were able to obtain new checkpoints with better ICL performance. As for now, for all datasets (except `city-roads-M`) there is at least one GraphPFN checkpoint that surpasses or nearly matches the performance of G2T-LimiX (ICL). We believe these results show the potential of GraphPFN.
> > >
> > > Second, regarding systematic analysis, we believe that it can be hard to obtain reliable analysis results using only 13 datasets. However, as we write in our paper, we hypothesize that poor performance on certain datasets (i.e., `hm-prices`, `avazu-ctr` and `twitch-views`) was caused by lack of big and dense enough graphs in our prior. We were able to obtain preliminary evidence for this hypothesis. Specifically, we finetuned the pretrained model on graphs with larger number of nodes and higher average density, and it significantly improved performance on certain datasets. Please refer to our general comment with additional experiments for the details.
> > >
> > > > **W7**. Presentation: The overall presentation has many problems. 1) Captions are sloppy and not informative. 2) Some terms are not defined such as "OOM" even though I understand what is stands for but it should be clearly defined.
> > >
> > > Thank you for your comment! We have updated the captions for Tables 2 and 3 with explanations for OOM and additional information. Please let us know if this addresses your concern or if you have any additional suggestions.
> > >
> > > > **W8**. More importantly, while the manuscript claims to focus on graph foundation models, it only evaluates node-level tasks, which makes its scope far too narrow.
> > >
> > > Let us note that some previous works like TS-GNN, GCOPE and G2T-FM are also suitable only for node-level tasks and yet they position themselves as graph foundation models. As shown in G2T-FM, many previously developed GFMs struggle to generalize to arbitrary node-level tasks from different domains. Thus, this task is already challenging enough. According to our evaluation, GraphPFN can successfully work with various graphs having different heterogeneous node features and under realistic, non-fewshot splits. We also refer to our general response where we elaborate on why we believe GraphPFN is an important step for GFMs.
> > >
> > > ***
> > >
> > > [1] Classic GNNs are Strong Baselines: Reassessing GNNs for Node Classification, NeurIPS 2024.
> > >
> > > [2] A critical look at the evaluation of GNNs under heterophily: Are we really making progress? ICLR 2023.

---

> > > > ### Comment · Reviewer_X1vi · 2025-11-26
> > > >
> > > > Thank you for your response, however my concerns remain valid after authors' rebuttal. Furthermore, I would strongly suggest authors to incorporate these necessary changes in their revised manuscript. I will raise the score to 4.

---

> > > > > ### Author Response · Authors · 2025-11-28
> > > > >
> > > > > Dear Reviewer X1vi,
> > > > >
> > > > > Thank you very much for your response and for taking the time to review our work.
> > > > >
> > > > > We have recently finished benchmarking time and memory efficiency, please see the results in [this comment](https://openreview.net/forum?id=BLJ5DsJ0i6&noteId=g5ZRzaYx2x). We have also updated the paper to address the question of Reviewer P18Y regarding some clarity issues in the abstract and introduction. We plan to add the new empirical results to the paper after the discussion with the reviewers, so we can properly address all the comments that may come up during the discussion. However, please note that the complete results of these experiments together with their discussion is now listed on Openreview and we are open to any questions or further discussions about them.
> > > > >
> > > > > We would greatly appreciate it if you could let us know which of your concerns remain unaddressed. This will help us improve our manuscript further.
> > > > >
> > > > > Best regards,
> > > > >
> > > > > Authors

---

### Official Review · Reviewer_PBKF · 2025-10-30

**Soundness:** 3
**Presentation:** 3
**Contribution:** 2
**Rating:** 6
**Confidence:** 3

**Summary:**

The authors propose GraphPFN, a method for adapting pre-trained tabular foundation models (TFM) to graph data by augmenting a TFM with a message-passing layer, and pre-train it on a large variety of synthetic graphs with causal, structurally-correlated node features. They demonstrate that GraphPFNs perform largely on par with or better than G2T-FMs across tasks in both in-context learning and fine-tuning setting, while avoiding expensive pre-processing required by G2T-FMs.

**Strengths:**

1. The core idea is interesting and potentially impactful. Leveraging pre-trained tabular foundation models (TFM) for GFMs is a sensible idea.
2. The integration of the graph adapters is a smart way of injecting structural inductive biases into TFMs effectively, without resorting to costly hand-crafted features (the avenue taken by G2T-FMs).
3. The results are overall convincing to the extent that the resulting models can compete with G2T-FMs and GNNs.
4. The paper is well-written, despite requiring substantial background on several areas the bare minimum for the background is mostly provided, and the overall presentation is clear and coherent.

**Weaknesses:**

1. The graph structure generation process seems mostly guided by an “eye test” on what “realistic” graphs _should_ look like. While I appreciate the authors’ efforts to go beyond simple synthetic graphs (SBM/ER/BA graphs etc.) and provide a generation algorithm that outputs a rather rich distribution of graphs, I’m not fully convinced that pre-training only on graphs from a single family of distributions is sufficient to capture real-world graph complexity. The authors clearly acknowledge this in section 5.2 and Appendix A (which I appreciate), where this phenomenon is reflected in the limited performance of GraphPFN (ICL) on certain datasets.
2. GraphPFN results are largely _on par with_, rather than consistently beating G2T-FMs across the benchmarks tested. I acknowledge that G2T-FMs typically perform better on downstream tasks where the datasets fit the graph prior better, however. Given comparable performance with G2T-FMs, I think the advantage of GraphPFN is the lack of reliance on hand-crafted features, and as a (likely result) faster evaluation, though the authors have not conducted any experiments (see Suggestion 1).
3. There are several issues with the evaluation procedures:
   - The GNN benchmarks are not provided with LapPE features, unlike GraphPFN,  resulting in an unfair comparison (see Suggestion 2).
   - Another factor that prevents fair comparison is that no information is provided on model sizes and training/pre-processing/inference efficiency. This is especially crucual in evaluating GraphPFN since it’s clear that the model is quite large, taking 6 days with eight A100s to pre-train _only_ the graph adapter layers, and leading to OOM errors in fine-tuning on certain datasets. When discussing foundation models parameter efficiency is an important metric to pay attention to, and it seems in this paper this dimension is wholly omitted (see Suggestion 1).

**Questions:**

**Questions:**
1. During pre-training, all layers except the graph adapters are frozen. Was a non-frozen model considered, and would there be there any potential benefits in pre-training the whole model?
2. Given the reliance of GraphPFN on synthetic graphs in pre-training despite the need for a realistic graph prior, can we not consider leveraging existing graph datasets either instead of or alongside synthetic graphs?
3. How does GraphPFN (with/without LapPE) compare with G2T-FMs in terms of any pre-processing and inference speed? I would expect GraphPFN (without LapPE) to perform inference faster since they don’t need to precompute encodings or structural statistics.

**Suggestions:**
1. More information regarding parameter counts and training/pre-processing/inference efficiency is required to better compare the GraphPFN and the baseline models (particularly the G2T-FMs) in a fair manner. I would imagine this would also help the authors make the point that not relying on hand-crafted features makes GraphPFN faster than G2T-FMs (assuming this holds; if not, please explain why).
2. Considering that some GNN benchmarks already outperform GraphPFN (ICL, without LapPE) on some datasets (`hm-prices`, `city-roads-M`, `questions`), GNN results with LapPE would provide a fairer comparison against stronger benchmarks; I suggest the authors provide them as well. Note that I am aware that the GraphPFN (ICL) results are with no additional training on the downstream dataset, so comparable results against GNNs are already impressive in their own right.
3. Referring to the UniMP model (Shi et al., 2021) as GT is confusing, given that acronym’s use for _graph transformers_ which typically leverage global attention over the graph. A minor suggestion is to use the UniMP convention.

**Conclusion:** I am very on the fence with this paper; the proposed idea is nice and seems to be implemented well. However, it seems to me the authors have failed to capitalize on the proposed method’s key advantage over G2T-FMs in not relying on costly hand-crafted features, and with performance against the baseline models being rather on par rather than a clear win, it’s hard to evaluate the true potential impact of the paper. I therefore recommend a tentative acceptance, and encourage the authors address the aforementioned concerns to arrive at a stronger paper.

---

> ### Author Response · Authors · 2025-11-21
> **Official Comment (Part 1/3)**
>
> Thank you for your review, insightful suggestions and support of our work. We address your questions and concerns below. Please also see the general response where we elaborate on why we believe GraphPFN is an important step for GFMs.
>
> > **W1**. The graph structure generation process seems mostly guided by an “eye test” on what “realistic” graphs should look like. While I appreciate the authors’ efforts to go beyond simple synthetic graphs (SBM/ER/BA graphs etc.) and provide a generation algorithm that outputs a rather rich distribution of graphs, I’m not fully convinced that pre-training only on graphs from a single family of distributions is sufficient to capture real-world graph complexity. The authors clearly acknowledge this in section 5.2 and Appendix A (which I appreciate), where this phenomenon is reflected in the limited performance of GraphPFN (ICL) on certain datasets.
>
> Thank you for your thoughtful comment!
>
> We agree that mixing distributions of different graph families can make a model more robust. It is reasonable to conduct such an experiment, but due to their high cost, we cannot be sure that we will manage to do this during the discussion period.
>
> As for now, we have certain updates regarding the graph prior that you may find interesting: 1) finetuning on dense graphs improves performance on certain datasets (see our general comment with additional experiments for details); 2) training solely on Erdős–Rényi graphs leads to significant drops in performance (see our response to Reviewer P18Y); 3) we conducted analysis of several properties of datasets from our prior and compared them with the properties of real-world datasets (see our response to the Reviewer X1vi for details).
>
> > **W2**. GraphPFN results are largely on par with, rather than consistently beating G2T-FMs across the benchmarks tested. I acknowledge that G2T-FMs typically perform better on downstream tasks where the datasets fit the graph prior better, however. Given comparable performance with G2T-FMs, I think the advantage of GraphPFN is the lack of reliance on hand-crafted features, and as a (likely result) faster evaluation, though the authors have not conducted any experiments (see Suggestion 1).
>
> > **W3.2**. Another factor that prevents fair comparison is that no information is provided on model sizes and training/pre-processing/inference efficiency. This is especially crucial in evaluating GraphPFN since it’s clear that the model is quite large, taking 6 days with eight A100s to pre-train only the graph adapter layers, and leading to OOM errors in fine-tuning on certain datasets. When discussing foundation models parameter efficiency is an important metric to pay attention to, and it seems in this paper this dimension is wholly omitted (see Suggestion 1).
>
> > **Q3**. How does GraphPFN (with/without LapPE) compare with G2T-FMs in terms of any pre-processing and inference speed? I would expect GraphPFN (without LapPE) to perform inference faster since they don’t need to precompute encodings or structural statistics.
>
> > **S1**. More information regarding parameter counts and training/pre-processing/inference efficiency is required to better compare the GraphPFN and the baseline models (particularly the G2T-FMs) in a fair manner. I would imagine this would also help the authors make the point that not relying on hand-crafted features makes GraphPFN faster than G2T-FMs (assuming this holds; if not, please explain why).
>
> We are currently benchmarking our models and will provide results when they are ready. However, please note that GraphPFN has 12 message passing layers, while G2T-FM does not have message passing layers at all. At the same time, besides possibly costly LapPE and PageRank, G2T-FM also exploits NFA, which can notably increase the input dimension due to one-hot encoding of categorical features. Thus, the comparison between GraphPFN and its counterparts in terms of inference speed can depend on the considered graph dataset.
>
> However, we also would like to note that our primary goal in this work is not to surpass G2T-FM or GNNs in terms of efficiency, but to achieve improvements in predictive performance. The results in our paper confirm that we manage to achieve that in most cases, and sometimes the performance gains are quite significant. We attribute these improvements to the fact that GraphPFN is not limited to 1-hop feature aggregations and can capture graph-specific patterns beyond hand-crafted features in G2T-FM. It is also pretrained on graph datasets, which enables it to capture more graph-specific dependencies in general.
>
> Moreover, please see our general comment in which we share additional experiments with GraphPFN pretraining: with some improvements of the learning procedure we were able to obtain new checkpoints with better ICL performance.

---

> > ### Author Response · Authors · 2025-11-21
> > **Official Comment (Part 2/3)**
> >
> > > **W3.1**. The GNN benchmarks are not provided with LapPE features, unlike GraphPFN, resulting in an unfair comparison (see Suggestion 2).
> >
> > > **S2**. Considering that some GNN benchmarks already outperform GraphPFN (ICL, without LapPE) on some datasets (hm-prices, city-roads-M, questions), GNN results with LapPE would provide a fairer comparison against stronger benchmarks; I suggest the authors provide them as well. Note that I am aware that the GraphPFN (ICL) results are with no additional training on the downstream dataset, so comparable results against GNNs are already impressive in their own right.
> >
> > Thank you for this thoughtful suggestion!
> >
> > Below, we provide the results of comparison of our new GraphPFN checkpoint (see our general comment with additional experiments for details) with G2T-LimiX and GNNs with enhanced features (i.e., provided all the features available to G2T-FM, namely, node degree, PageRank, LapPE, NFA and PEARL). As one can see, GraphPFN-1.1 (ICL) is highly competitive, particularly in terms of average rank, even against GNNs with enhanced features.
> >
> > |                            | pubmed       | facebook     | amazon-ratings | questions    | wiki-cs      | tolokers-2   | city-reviews | artnet-exp   | hm-prices    | avazu-ctr    | city-roads-M | twitch-views | artnet-views | AR   |
> > | :------------------------- | :----------- | :----------- | :------------- | :----------- | :----------- | :----------- | :----------- | :----------- | :----------- | :----------- | :----------- | :----------- | :----------- | :--- |
> > | GCN (M)                    | 84.57 ± 0.13 | 92.35 ± 0.12 | 39.46 ± 1.95   | 14.65 ± 0.58 | 81.34 ± 0.19 | 58.71 ± 0.45 | 77.07 ± 0.27 | 43.44 ± 0.32 | 70.73 ± 0.26 | 31.10 ± 0.22 | 57.91 ± 0.22 | 77.11 ± 0.09 | 56.14 ± 0.24 | 4.92 |
> > | GraphSAGE (M)              | 84.76 ± 0.18 | 92.30 ± 0.07 | 41.40 ± 0.47   | 14.13 ± 0.77 | 81.19 ± 0.31 | 59.59 ± 0.51 | 77.95 ± 0.09 | 44.31 ± 0.53 | 70.50 ± 0.47 | 31.51 ± 0.41 | 59.66 ± 0.09 | 75.93 ± 0.19 | 55.39 ± 0.32 | 4.69 |
> > | GAT (M)                    | 83.70 ± 0.45 | 92.66 ± 0.19 | 41.69 ± 0.27   | 18.50 ± 0.68 | 81.39 ± 0.38 | 57.76 ± 0.70 | 77.47 ± 0.14 | 44.36 ± 0.50 | 72.46 ± 0.49 | 31.97 ± 0.23 | 59.57 ± 0.43 | 77.04 ± 0.08 | 56.51 ± 0.35 | 3.69 |
> > | GT (M)                     | 83.89 ± 0.28 | 92.75 ± 0.15 | 41.65 ± 0.42   | 16.14 ± 0.87 | 81.08 ± 0.38 | 58.79 ± 0.76 | 76.43 ± 0.10 | 43.03 ± 0.60 | 71.84 ± 0.64 | 29.86 ± 0.67 | 59.85 ± 0.41 | 76.15 ± 0.11 | 56.39 ± 0.31 | 4.62 |
> > | G2T-LimiX (ICL)            | 88.96 ± 0.18 | 91.29 ± 0.14 | 44.10 ± 0.16   | 15.31 ± 0.77 | 79.99 ± 0.28 | 61.48 ± 0.30 | 77.72 ± 0.54 | 48.43 ± 0.18 | 74.96 ± 0.06 | 32.39 ± 0.14 | 64.53 ± 0.07 | 71.08 ± 0.07 | 60.95 ± 0.10 | 2.85 |
> > | GraphPFN-1.1 (ICL)         | 89.53 ± 0.03 | 91.21 ± 0.01 | 43.88 ± 0.03   | 18.67 ± 0.04 | 79.48 ± 0.05 | 60.01 ± 0.03 | 79.31 ± 0.01 | 50.30 ± 0.01 | 71.45 ± 0.02 | 30.07 ± 0.01 | 61.86 ± 0.03 | 62.37 ± 0.04 | 61.70 ± 0.01 | 3.46 |
> > | GraphPFN-1.1 + LapPE (ICL) | 89.40 ± 0.15 | 91.00 ± 0.23 | 43.32 ± 0.15   | 12.48 ± 0.37 | 79.82 ± 0.25 | 61.47 ± 0.17 | 79.56 ± 0.06 | 50.26 ± 0.05 | 71.57 ± 0.06 | 30.83 ± 0.22 | 61.80 ± 0.15 | 69.08 ± 0.34 | 62.93 ± 0.04 | 3.77 |

---

> > > ### Author Response · Authors · 2025-11-21
> > > **Official Comment (Part 3/3)**
> > >
> > > > **Q1**. During pre-training, all layers except the graph adapters are frozen. Was a non-frozen model considered, and would there be there any potential benefits in pre-training the whole model?
> > >
> > > We believe that, during the first steps of pretraining, the TFM backbone should be frozen, since it already brings various strong capabilities, such as processing diverse heterogeneous node features and making predictions via in-context learning, so we want to preserve them as much as possible. However, we also believe that it can be unfrozen later to improve the model adaptation to graph-specific tasks. It is reasonable to conduct such an experiment, but due to the high computational cost, we cannot promise that we will manage to do this during the discussion period.
> > >
> > > > **Q2**. Given the reliance of GraphPFN on synthetic graphs in pre-training despite the need for a realistic graph prior, can we not consider leveraging existing graph datasets either instead of or alongside synthetic graphs?
> > >
> > > Indeed, in the tabular domain, there are some cases of using only real data for pretraining a foundation model, such as the TabDPT [1] that trains a model using various self-supervised tasks on a variety of real tabular datasets. However, according to the results on TabArena [2, 3], it is outperformed by TabPFN in the ICL mode, which suggests that properly designed synthetic prior distribution can provide more useful knowledge than some fixed set of real datasets. Moreover, there can not always be enough high-quality real-world datasets for pretraining and, in the graph domain, this problem is particularly pressing. Because of that, pretraining using synthetic data appears to be a reasonable and sometimes even necessary decision.
> > >
> > > Nevertheless, as shown by Real-TabPFN [4], one can pretrain a foundation model on synthetic data and then finetune it on a collection of high-quality real-world datasets, and such an approach may provide even better results. So, in theory, GraphPFN can also be finetuned on a collection of real graphs, but we are not sure what amount of graph data is needed and whether the graph ML domain has enough high-quality datasets. Thus, we leave this for future research.
> > >
> > > ***
> > >
> > > [1] TabDPT: Scaling Tabular Foundation Models on Real Data, NeurIPS 2025
> > >
> > > [2] TabArena: A Living Benchmark for Machine Learning on Tabular Data, NeurIPS 2025
> > >
> > > [3] TabArena leaderboard at https://huggingface.co/spaces/TabArena/leaderboard
> > >
> > > [4] Real-TabPFN: Improving Tabular Foundation Models via Continued Pre-training With Real-World Data, ICML 2025

---

> > > > ### Author Response · Authors · 2025-11-28
> > > >
> > > > Dear Reviewer PBKF,
> > > >
> > > > We have recently finished benchmarking time and memory efficiency, please see the results in [this comment](https://openreview.net/forum?id=BLJ5DsJ0i6&noteId=g5ZRzaYx2x). If you have any other concerns remaining, please let us know, we would be happy to discuss them further.
> > > >
> > > > Best regards,
> > > >
> > > > Authors

---

### Official Review · Reviewer_P18Y · 2025-11-01

**Soundness:** 1
**Presentation:** 2
**Contribution:** 1
**Rating:** 2
**Confidence:** 4

**Summary:**

This paper introduces GraphPFN, a prior-data fitted network for node-level prediction that builds graph foundation models (GFMs) through synthetic pretraining. It generates diverse, realistic graphs using a hybrid of stochastic block models and preferential attachment, combined with graph-aware causal models for attributes and targets. By augmenting the tabular foundation model with attention-based neighborhood aggregation and training on these synthetic graphs, GraphPFN achieves strong performance.

**Strengths:**

1. This paper studies an interesting and practical research question of applying synthetic data in large-scale pretraining.
2. The proposed data generation pipeline seems reasonable.

**Weaknesses:**

1. The writing of this paper is confusing and the core contribution of the work is not well presented. In the introduction (line 066), the authors mentioned "but such models still depend heavily on hand-crafted features and, as a result, are limited in their ability to capture complex graph patterns." This challenge is addressed by the proposed message-passing layer which replaces the structural features. However, the motivation of utilizing synthetic data pretraining is not discussed, which has no clear connection to the challenges mentioned.
2. Efficiency concern: the proposed method builds on existing TFMs and requires post-training involving massive compute resources(8*A100*6days). At inference time, it involves dataset-specific fine-tuning to achieve the claimed SOTA performance. Given the lack of efficiency comparison, I doubt the real gains of the proposed method over baselines.
3. Misleading experimental results: Table 2 compares GraphPFN with baselines. However, the gains of GraphPFN are inconsistent, with OOM issues which were not observed in baselines. Since a core edge of GraphPFN is that it does not rely on hand-crafted features, it is not reasonble to include the results for the LapPE enhanced version. As a result, the original version of the proposed method is less competitive, especially when compared with G2T.
4. The data generation process involves many hyperparameters. Though the authors claim that they can be sampled from some distribution, the details are missing and the method still heavily relies on trial-and-error, limiting its real-world applicability.

**Questions:**

1. What is the initial motivation of applying synthetic data pretraining? Can it be replaced by real-world data, or a mix of both?
2. How does the properties and amounts of pretraining data affect performance?

---

> ### Author Response · Authors · 2025-11-21
> **Official Comment (Part 1/2)**
>
> Thank you for your review. We address your questions and concerns below. Please also see the general response where we elaborate on why we believe GraphPFN is an important step for GFMs.
>
> > **W1**. The writing of this paper is confusing and the core contribution of the work is not well presented. In the introduction (line 066), the authors mentioned "but such models still depend heavily on hand-crafted features and, as a result, are limited in their ability to capture complex graph patterns." This challenge is addressed by the proposed message-passing layer which replaces the structural features. However, the motivation of utilizing synthetic data pretraining is not discussed, which has no clear connection to the challenges mentioned.
>
> > **Q1**. What is the initial motivation of applying synthetic data pretraining? Can it be replaced by real-world data, or a mix of both?
>
> As we state in our general response, we believe that the PFN framework offers a promising direction towards foundation models in the graph domain. Pretraining on large amounts of synthetic data is an essential component of the PFN framework and we follow this approach. We believe that utilizing synthetic data is well-suited for the graph ML domain since this field lacks a sufficient amount of real-world high-quality pretraining data compared to, e.g., CV and NLP. Additionally, we are motivated by the fact that the most successful current tabular foundation models are pretrained on synthetic data, and we believe that this success can be transferred to the graph domain. On the other hand, real-world datasets can potentially be used to further strengthen the model by finetuning it on a collection of real datasets, as shown in Real-TabPFN [1].
>
> Thank you for this comment! We will update the paper to better reflect our motivation during the discussion period.
>
> > **W2**. Efficiency concern: the proposed method builds on existing TFMs and requires post-training involving massive compute resources(8A1006days). At inference time, it involves dataset-specific fine-tuning to achieve the claimed SOTA performance. Given the lack of efficiency comparison, I doubt the real gains of the proposed method over baselines.
>
> We acknowledge that, in its current form, GraphPFN has issues with scalability and efficiency. Yet, as we write in our general response, we believe that now it is more important to demonstrate the applicability of this approach in principle, and we leave scaling to larger graphs to future research.
>
> > **W3**. Misleading experimental results: Table 2 compares GraphPFN with baselines. However, the gains of GraphPFN are inconsistent, with OOM issues which were not observed in baselines. Since a core edge of GraphPFN is that it does not rely on hand-crafted features, it is not reasonble to include the results for the LapPE enhanced version. As a result, the original version of the proposed method is less competitive, especially when compared with G2T.
>
> It is true that the current version of GraphPFN faces some issues with efficiency and scalability that are caused by certain limitations of the PFN framework. However, even the results that we have managed to obtain on most datasets with GraphPFN in either ICL or finetuning mode are already promising, since it often outperforms all possible variations of G2T and supervised task-specific GNNs. We also admit that using LapPE is not an elegant solution, and it would be great to work without it, but we think that it is still interesting to investigate how such additional structural encodings can improve the predictive performance of GraphPFN, since G2T-FM always receives them as input features. As can be seen, they help to achieve even better results on some datasets. Moreover, please see our general comment in which we share additional experiments with GraphPFN pretraining: with some improvements of the learning procedure we were able to obtain new checkpoints with better ICL performance.

---

> > ### Author Response · Authors · 2025-11-21
> > **Official Comment (Part 2/2)**
> >
> > > **W4**. The data generation process involves many hyperparameters. Though the authors claim that they can be sampled from some distribution, the details are missing and the method still heavily relies on trial-and-error, limiting its real-world applicability.
> >
> > We admit that tuning prior distribution for the PFN framework is a challenging empirical task that requires a large number of experiments and careful investigation of the effects caused by changing the properties of the prior distribution. However, given that PFNs serve as a very promising direction to develop foundation models in both tabular and graph domains, we believe that designing a prior distribution that accurately describes the properties of real-world tasks is one of the main problems that research community should now focus on to make new advancements in the GFM field. Our work makes the first step towards solving this problem.
> >
> > Also, could you please clarify what details are missing, so we can add them?
> >
> > > **Q2**. How does the properties and amounts of pretraining data affect performance?
> >
> > Please note that a detailed ablation of different properties in prior distribution can be very costly, as each particular change in the generation procedure may require a separate run of pretraining the whole GraphPFN model. As for now, the learning curves provided in our paper can serve as a proxy for the amounts of pretraining data, since the longer GraphPFN model pretraines, the more distinct synthetic datasets it observes. Regarding the properties of prior distribution, we have also experimented with replacing the proposed graph generation process with a simpler Erdős–Rényi (ER) model, as the most basic approach. The table below provides comparison of ICL performance of corresponding model (GraphPFN-ER) against default GraphPFN. One can clearly see that pretraining on ER graphs produces significantly worse results, justifying the usage of more complex random graph models.
> >
> > |                   | tolokers-2 | artnet-exp | city-roads-M | artnet-views |
> > | :---------------- | :--------- | :--------- | :----------- | :----------- |
> > | GraphPFN (ICL)    | 58.95      | 49.97      | 58.93        | 58.87        |
> > | GraphPFN-ER (ICL) | 45.43      | 42.93      | 58.06        | 43.87        |
> >
> > ***
> >
> > [1] Real-TabPFN: Improving Tabular Foundation Models via Continued Pre-training With Real-World Data, arXiv:2507.03971, 2025.

---

> > > ### Author Response · Authors · 2025-11-28
> > >
> > > Dear Reviewer P18Y,
> > >
> > > We have recently finished benchmarking time and memory efficiency, please see the results in [this comment](https://openreview.net/forum?id=BLJ5DsJ0i6&noteId=g5ZRzaYx2x). Also, we have updated the introduction and abstract in the new revision to clarify the positioning of our work, and in particular motivate pretraining on synthetic datasets. If you have any other questions or suggestions, please let us know.
> > >
> > > Best regards,
> > >
> > > Authors

---

### Author Response · Authors · 2025-11-21
**New Experimental Results**

Dear Reviewers,

In this general comment, we present some new experimental results that are relevant to the comments of several reviewers.

In the table below we show the results of new checkpoints that have better downstream performance. Here, GraphPFN is the checkpoint from the submission version and GraphPFN-1.1 is the updated version. Most important changes include adding EMA weight averaging during pretraining for improving stability and updating hyperparameters of the prior to cover larger and more dense graphs. As one can see, the overall results are better now, with particularly noticeable improvements on `hm-prices`, `avazu-ctr`, `twitch-views` and `artnet-views`.

We also experimented with finetuning GraphPFN-1.1 on larger graphs with even higher density, and the corresponding checkpoint is called GraphPFN-1.1-dense. It even further improves the results on `hm-prices` and `twitch-views`, supporting our hypothesis that poor performance on these datasets was caused by lack of dense enough graphs in the prior.

|                                   | pubmed       | facebook     | amazon-ratings | questions    | wiki-cs      | tolokers-2   | city-reviews | artnet-exp   | hm-prices    | avazu-ctr    | city-roads-M | twitch-views | artnet-views |
| :-------------------------------- | :----------- | :----------- | :------------- | :----------- | :----------- | :----------- | :----------- | :----------- | :----------- | :----------- | :----------- | :----------- | :----------- |
| G2T-LimiX (ICL)                   | 88.96 ± 0.18 | 91.29 ± 0.14 | 44.10 ± 0.16   | 15.31 ± 0.77 | 79.99 ± 0.28 | 61.48 ± 0.30 | 77.72 ± 0.54 | 48.43 ± 0.18 | 74.96 ± 0.06 | 32.39 ± 0.14 | 64.53 ± 0.07 | 71.08 ± 0.07 | 60.95 ± 0.10 |
| GraphPFN (ICL)                    | 89.76 ± 0.02 | 90.23 ± 0.03 | 43.71 ± 0.03   | 12.30 ± 0.05 | 77.29 ± 0.03 | 58.95 ± 0.02 | 79.52 ± 0.01 | 49.97 ± 0.01 | 68.02 ± 0.02 | 21.67 ± 0.05 | 58.93 ± 0.02 | 59.95 ± 0.08 | 58.87 ± 0.01 |
| GraphPFN  + LapPE (ICL)           | 89.41 ± 0.09 | 91.45 ± 0.16 | 42.64 ± 0.22   | 9.62 ± 0.16  | 79.13 ± 0.37 | 61.18 ± 0.11 | 79.73 ± 0.08 | 49.68 ± 0.04 | 68.56 ± 0.10 | 23.33 ± 0.47 | 58.61 ± 0.41 | 63.66 ± 0.77 | 60.31 ± 0.09 |
| GraphPFN-1.1 (ICL)                | 89.53 ± 0.03 | 91.21 ± 0.01 | 43.88 ± 0.03   | 18.67 ± 0.04 | 79.48 ± 0.05 | 60.01 ± 0.03 | 79.31 ± 0.01 | 50.30 ± 0.01 | 71.45 ± 0.02 | 30.07 ± 0.01 | 61.86 ± 0.03 | 62.37 ± 0.04 | 61.70 ± 0.01 |
| GraphPFN-1.1 + LapPE (ICL)        | 89.40 ± 0.15 | 91.00 ± 0.23 | 43.32 ± 0.15   | 12.48 ± 0.37 | 79.82 ± 0.25 | 61.47 ± 0.17 | 79.56 ± 0.06 | 50.26 ± 0.05 | 71.57 ± 0.06 | 30.83 ± 0.22 | 61.80 ± 0.15 | 69.08 ± 0.34 | 62.93 ± 0.04 |
| GraphPFN-1.1-dense (ICL)          | 89.06 ± 0.03 | 90.83 ± 0.03 | 42.24 ± 0.04   | 15.93 ± 0.04 | 79.14 ± 0.02 | 60.43 ± 0.03 | 78.00 ± 0.01 | 51.46 ± 0.01 | 74.09 ± 0.02 | 29.63 ± 0.01 | 60.27 ± 0.04 | 68.07 ± 0.06 | 61.64 ± 0.01 |
| GraphPFN-1.1-dense  + LapPE (ICL) | 88.51 ± 0.09 | 90.02 ± 0.30 | 42.18 ± 0.15   | 10.60 ± 0.63 | 80.13 ± 0.19 | 61.35 ± 0.21 | 77.97 ± 0.15 | 51.87 ± 0.07 | 74.18 ± 0.05 | 30.90 ± 0.32 | 60.55 ± 0.22 | 71.60 ± 0.28 | 62.26 ± 0.09 |

Best regards,

Authors

---

> ### Author Response · Authors · 2025-11-26
> **Measuring time and memory efficiency**
>
> Dear Reviewers,
>
> We are currently working on incorporating all the discussed changes to the paper. For now, we would like to share some additional experiments on efficiency.
>
> As we mention in our general response, we acknowledge potential limitations of GraphPFN in terms of efficiency and scalability, yet we believe the work to be valuable for the GFM field. However, for completeness, below we provide the results of benchmarking GraphPFN, G2T-FM and GCN in terms of time and memory efficiency.
>
> For each dataset, all methods were evaluated in exactly the same environment on the same machine, making the comparison fair. For all methods, we use the optimal hyperparameters chosen during our main experiments, and only make one evaluation with a single random seed.
>
> The table below contains peak memory consumption measured in GiB. One can clearly see that the memory consumption of GraphPFN is noticeably higher than those of GCN and G2T-FM.
>
> |           | tolokers-2 | artnet-exp | hm-prices | city-roads-M | artnet-views | city-reviews | avazu-ctr | twitch-views |
> | :-------- | :--------- | :--------- | :-------- | :----------- | :----------- | :----------- | :-------- | :----------- |
> | GCN       |       0.56 |        2.1 |      2.53 |         2.17 |         1.93 | 5.85         | 3.69      | 6.59         |
> | G2T-LimiX |        3.5 |      35.61 |      46.7 |        15.21 |        20.02 | 49.48        | 36.83     | 40.17        |
> | GraphPFN  |       4.14 |      66.84 |     50.86 |        22.09 |        45.85 | N/A          | N/A       | N/A          |
>
>
> We also measure time (in seconds) required to preprocess a dataset (“simple” means that we only load it from disk and transform to desired format, and in “G2T-FM” preprocessing we also compute LapPE, node degree, PageRank and NFA used in G2T-FM), to train (or finetune) a model, and to infer a model on a dataset.
>
> |                       | tolokers-2 | artnet-exp | hm-prices | city-roads-M | artnet-views | city-reviews | avazu-ctr | twitch-views |
> | :-------------------- | :--------- | :--------- | :-------- | :----------- | :----------- | :----------- | :-------- | :----------- |
> | Preprocessing: simple |       0.45 |       1.46 |     11.16 |         0.35 |         1.08 | 2.15         | 14.93     | 7.17         |
> | Preprocessing: G2T-FM |       1.16 |       5.14 |     27.88 |        42.49 |         3.27 | 30.47        | 54.92     | 14.56        |
> | Train: GCN            |      25.77 |      41.58 |      77.9 |        22.71 |        19.53 | 85.66        | 32.90     | 327.46       |
> | Finetune: G2T-LimiX   |     139.52 |     832.07 |    1123.4 |        225.9 |       457.12 | 1688.98      | 1013.35   | 1680.59      |
> | Finetune: GraphPFN    |      85.59 |     828.03 |   5426.91 |        339.8 |       690.36 | N/A          | N/A       | N/A          |
> | Inference: G2T-LimiX  |       0.87 |      16.39 |     20.77 |         3.58 |         5.54 | 27.27        | 15.14     | 21.32        |
> | Inference: GraphPFN   |       0.44 |       3.48 |     12.01 |         1.16 |         2.44 | N/A          | N/A       | N/A          |
>
> From the table above, one can see that
> * Inference of GraphPFN is faster than training GNN from scratch;
> * Inference of GraphPFN is faster than those of G2T-LimiX (even without considering preprocessing overhead for G2T-LimiX);
> * Finetuning of both GraphPFN and G2T-LimiX is significantly longer compared to inference and preprocessing;
> * Finetuning of GraphPFN can be significantly longer compared to finetuning of G2T-LimiX.
>
> Finally, we would like to note that since efficiency was not the primary goal of our work, neither of the considered methods was fully optimized in terms of efficiency. Hence, some methods can benefit from further efficiency optimization more than others. However, we believe that the results above reflect the overall picture.
>
> Best regards,
>
> Authors

---

### Author Response · Authors · 2025-11-21
**General Response**

Dear Reviewers,

We thank you for your insightful feedback. We addressed the specific comments and questions in individual responses below. In this general response, we want to elaborate on why we believe GraphPFN is an important step for GFMs.

The development of GFMs, even those restricted to node-level tasks only, faces several fundamental challenges. First, **graphs do not constitute a single domain**, but rather serve as a way to represent data from different domains from social and web networks to road graphs. This makes it challenging to achieve knowledge transfer between different datasets. Second, there is a **lack of high-quality, large-scale, diverse graph datasets for pretraining GFMs**. In contrast to CV or NLP, the amount of available data in graph ML is orders of magnitude smaller.

In our opinion, **none of prior GFMs, except recently proposed G2T-FM, managed to convincingly address these challenges** for graphs with arbitrary node features, meaningful prediction tasks and realistic data splits. Most progress so far has been limited to either text-attributed graphs, where tasks can be unified via text encoders, or to few-shot scenarios, which are less practical for many real-world applications.

At the same time, **tabular ML faces similar challenges, and this domain has recently seen the emergence of successful foundation models**, such as TabPFNv2 and LimiX. These models demonstrate strong empirical performance and, in certain settings, outperform task-specific models trained from scratch. Most TFMs follow the PFN framework [1], and we believe that, as for now, **PFNs offer the most promising path toward foundation models for both tabular and graph domains**.

**Our position is supported by experimental results of PFNs in the tabular domain**, and we also have several hypotheses about why PFNs function well as foundation models. First, their in-context capabilities enable them to potentially capture patterns not only at the level of individual samples, but also at the level of the whole dataset, treating other samples as useful context. We suppose that this can be crucial for heterogeneous domains like graphs or tables. Second, given the insufficient amount of real-world pretraining data in comparison to CV and NLP, the use of synthetic datasets becomes a feasible and effective alternative. The flexibility of synthetic data also enables us to control the knowledge acquired by the foundation model.

However, **despite their success in the tabular domain, PFNs are not yet fully adopted to graphs**, and GraphPFN is the next step in this direction. While G2T-FM has shown that PFN-based TFMs can outperform both GNNs and prior GFMs, it relies on hand-crafted features to capture graph interactions and was not pretrained on graph data. In GraphPFN, we address these limitations by using learnable message passing layers and pretraining the model on synthetic datasets drawn from a carefully designed prior. To the best of our knowledge, GraphPFN is the first successful model pretrained in the PFN framework specifically for graph data.

Therefore, **the key contribution of our work is that it shows that the PFN framework can be adapted for graph learning tasks and show decent performance**. Although much work remains, including the design of a better graph prior distribution, the investigation of a more effective graph-specific PFN architecture, scaling to larger graphs and addressing link- and graph-level tasks, our results show a clear and promising direction forward. We hope that our work will support further development of GFMs and will inspire further research in this area.

Best regards,

Authors

---

### Author Response · Authors · 2025-12-03
**Rebuttal summary**

Dear Reviewers and Area Chair,

We would like to thank the reviewers for their detailed feedback and insightful suggestions that helped us clarify key points and strengthen our work. Unfortunately, current rules do not allow the reviewers to continue the discussion. However, we believe that we have addressed the reviewers' concerns. Below, we summarize the reviews and our responses.

First of all, we would like to point to our [general response](https://openreview.net/forum?id=BLJ5DsJ0i6&noteId=l2nhuiwXEv), where we share our view on current progress and challenges in the field of graph foundation models. Here we elaborate on why we believe that PFNs offer a promising path towards building GFMs and GraphPFN makes an important step in adapting PFNs to graph tasks by pretraining a PFN-based model specifically for graph data.

Now, let us summarize strengths and weaknesses mentioned by the reviewers.

**Strengths:**
- **Architecture:**
	- `PBKF`: "The integration of the graph adapters is a smart way of injecting structural inductive biases into TFMs effectively, without resorting to costly hand-crafted features"
	- `X1vi`: "Rather than training a graph model from scratch, the authors initialize from LimiX and add sparse message-passing adapters that respect the per-feature tokenization. The motivation seems reasonable since it lets the model learn what to aggregate rather than relying entirely on hand-crafted features like G2T-FM does."
	- `4qnR`: The paper proposes a clear architectural idea: lightweight graph adapters (masked attention over adjacency) added to each LimiX block; retains PFN ICL while enabling learned structural reasoning.
- **Prior design:**
	- `P18Y`: "The proposed data generation pipeline seems reasonable"
	- `X1vi`: "The proposed method combines multiple degree-corrected SBMs to address the "too clean" cluster problem in single-SBM generations, and the preferential attachment augmentation naturally to create the core-periphery structure in social and information networks, which sounds interesting"
	- `4qnR`: "The paper combines multi-SBM with preferential attachment to mimic community + core-periphery; attributes from graph-aware SCM (MLP/GNN mix, optional LapPE)"

**Weaknesses:**
- **Scalability:** We acknowledge potential limitations of GraphPFN in terms of scalability. However, these limitations are inherent to all PFN-based models, and we hope that future progress in this area will allow us to overcome these issues. As for now, GraphPFN serves as a proof of concept that one can train a PFN-based model specifically for graph data.
- **Weak ICL performance on some datasets:** We were able to obtain better checkpoints of GraphPFN, see [our comment "New Experimental Results”](https://openreview.net/forum?id=BLJ5DsJ0i6&noteId=Zsj8bbQdDT) for details. In particular, we were able to drastically improve performance on the datasets with poor ICL performance in the initial submission.
- **Clarity of presentation:** We have updated our paper to clarify the motivation of our work. We again thank the Reviewer P18Y for highlighting this.
- **Prior analysis:** We have conducted quantitative analysis of the proposed prior, see our [response to the Reviewer X1vi](https://openreview.net/forum?id=BLJ5DsJ0i6&noteId=EPW0OAb4f7) for details.
- **Ablation:** Due to the high cost of GraphPFN pretraining, we were unable to ablate all possible design choices. In [this response to the Reviewer 4qnR](https://openreview.net/forum?id=BLJ5DsJ0i6&noteId=uSWx0SI23O) we provided results of ablating MGM and StrucFeat components. For other design choices, we provided our motivation to select them.

Overall, we believe that the reviewers' concerns while being valid do not override the importance of our main contribution, which is definitely valuable for the GNN field given the strong empirical results and the potential of the PFN framework for graph tasks.

Best regards,
Authors

---

### Meta-Review · Area_Chair_YcRV · 2026-01-07

**Summary:**

The paper proposes GraphPFN, a prior-data fitted network for node-level prediction that builds graph foundation models (GFMs) through synthetic pretraining.  The authors demonstrated that GraphPFNs perform on par or better with G2T-FMs across tasks in in-context learning and fine-tuning.

The majority of the reviews lean towards reject, and the positive one still has major concerns that are in line with the other reviewers.  The main concerns are the scalability and efficiency issues of the proposal, as well as the unconvincing results that are limited by the ablations and heuristics used.  Given these severe issues, I recommend the rejection of the paper.


Strengths:
- Interesting and practical research question leveraging pre-trained tabular foundation models (TFMs) for GFM
- The use of graph adapters is an interesting way of injecting structural inductive biases into TFMs
- Reasonable pipeline data generation


Weaknesses:
- Lack of scalability and efficiency comparisons
- Inconsistent improvements in the different experiments
- Presentations issues for the main contribution
- The graph generation process is mostly guided by heuristics

**Reviewer Concerns:**

Reviewer P18Y raised concerns about the presentation of the contributions, lack of efficiency comparison, misleading experimental results since the improvements are not consistent, and the missing details for the data generation process.  The authors replied to the reviewer concerns.  However, there were still issues with the efficiency limitations and the tuning of the prior distributions for the PFN framework.

Reviewer PBKF raised concerns about the heuristics in the graph structure generation process, limited improvements against G2T-FMs across benchmarks, and unfair evaluation procedures (not using LapPE features, lack of information in hyperparameters), and lack of real graph datasets as realistic priors.  The authors replied to the reviewer's concerns and provided additional experiments.  There are still issues with the evaluation of unfrozen backbones.

Reviewer X1vi raised concerns about the scalability and the claims of a general model ("Foundation Model"), the prior design since pretraining on synthetic graphs lacks support given the systematic failures, lack of ablations on several key decisions (e.g., individual contribution of the masked graph modeling objective vs. supervised PFN loss), and lack of quantitative validation that synthetic graphs match real-world distributions.  The authors replied to the comments and provided additional results.  However, there are still issues w.r.t. missing ablations to justify design choices, as well as ad-hoc hyperparameters per dataset, and limited evaluations (node-level tasks only).

Reviewer 4qnR raised concerns about the efficiency being a bottleneck, the narrow tasks, the limited baselines, the application of the proposal to other graph-related tasks, and missing ablations for different graph processing as well as other method components.  The authors provided answers to the reviewer's concerns.  However, the proposal is still limited to nodes-only tasks, and overall the main concern are the unconvincing results.

**Reviewer Scores:**

Reviewer P18Y recommended a reject.  Despite the authors efforts in the rebuttal, the issues with the scalability and efficiency, as well as the issues with the priors remain.

Reviewer PBKF recommended a weak accept.  While the authors replied and provided additional experiments, there were still issues regarding the performance improvement and whether unfrozen backbones will improve the results as well as real data.  The authors did not provide results only conjectures.

Reviewer X1vi recommends a rejection.  Despite the authors responses, there were major issues that are not solved such as missing ablations and ad-hoc hyperparameter searches per dataset, scalability issues.  The reviewer acknowledge the authors' effort and raised the score to weak reject but still recommends a rejection.

Reviewer 4qnR recommended a rejection.  Given the open concerns regarding unconvincing results and the issues with efficiency, the score is still leaning towards a reject.

---

### Decision · Program_Chairs · 2026-01-26

Reject